# The impact of universal recycling on the evolution of economic diversity

**Shubo (Gabriel) Xu** **\*, Charles S. Peskin**

Courant Institute of Mathematical Sciences, New York University, New York, NY, United States of America

\* sx663@nyu.edu

**Data Availability Statement:** All relevant data and Matlab codes are uploaded to the DANS repository (https://urldefense.proofpoint.com/v2/url?u=https-3A__doi.org_10.17026_dans-2Dz3k-2D95ya&d=DwlGaQ&c=slrrB7dE8n7gBJbeO0g-IQ&r=UlRc71lOrJioUEm0izKH_w&m=xTChvGH5-yhi-

## Abstract

Based on von Neumann's model of an economy characterized by processes and goods, we add to that model a component representing capital equipment. We assume that the need for capital equipment by any process is proportional to the rate at which that process is running, and therefore an increase in rate requires that capital equipment be purchased, whereas a decrease in rate allows capital equipment to be sold. We thereby construct a continuous-time dynamical model, which we use to investigate the evolution of economic diversity under two price equilibrium scenarios: the first with non-negative prices and non-positive excess demands; the second with enforced market clearing and with prices allowed to be negative. The second scenario represents an economy in which recycling is required, so that excess supply cannot be discarded. We prove that at any time during the progression of the model economy, the solution to each of the two price equilibrium problems exists, and that non-uniqueness of the solution, if any, does not affect the development of the model economy. We compare matched model economies under the two scenarios by simulating their respective evolutions. In each case, the model economy experiences a process of selection and matures to a state of balanced growth, with a higher growth rate when excess supply is discarded, but with greater economic diversity with enforced recycling. The robustness of these qualitative results is demonstrated by repeated trials of simulations on matched pairs of model economies with different randomly chosen parameters.

## Introduction

In this paper, we reformulate a model of von Neumann [1] so that it includes capital goods, and we use the reformulated model to carry out numerical simulations of economic evolution. We investigate the model's dynamics to conduct a comparative study of economic growth, diversity and price equilibrium under two different assumptions related to market clearing.

In the first version of our model, we assume, along with von Neumann, that excess demand cannot be positive but that negative excess demand (i.e., positive excess supply) is allowed, and that any good in excess supply becomes a free good. In this scenario, as in most economic theories, prices are constrained to be non-negative.

What this set of assumptions overlooks is that the free disposal of free goods may be harmful to the environment, and that antipollution laws may be passed to prevent such harm. In

R8OlKzV9dFmjrdV_
8hKC72eRbEaWKSex0DBEZu3Z7CDjH93GlOi&s=
NuxhBb5xql3FtDVYeMsT99K9M2rFT-bV3Jp_
T5MITYc&e=).

**Funding:** The author(s) received no specific funding for this work.

**Competing interests:** The authors have declared that no competing interests exist.

this situation, a firm may need to pay another firm to take the unwanted byproducts of production off its hands. A familiar example of this in everyday life is when a household pays 1–-800-GOT-JUNK to remove its unwanted household goods. The distinctive feature of this kind of transaction is that goods and money are transferred in the same direction, and this implies that the goods in question have a negative price. (There is also a *service* involved in the transaction, and the service has a positive price. Let $P_s > 0$ be the price of the service of removing one unit of the unwanted good, and let $P_g < 0$ be the price of the unwanted good itself. Then the amount of money that the household pays per unit of removal of the good is $P_s - P_g$, and this is greater than the price of the service alone, since the good is unwanted and its price is negative. Note that the formula $P_s - P_g$ makes sense also for a good with positive price. For example, if the price of the good is equal to the price of the service of removing that good, then no money needs to change hands, and if the price of the good is greater than the price of the service of removing it, then the firm that is doing the removal has to pay for the privilege, since it gets a more valuable good in exchange.) Indeed, in early 2020, there was so little demand for oil as a result of the current pandemic that oil prices briefly became negative [2].

In the second version of our model, we accordingly assume perfect market clearing for all goods, and we allow the prices of goods to become negative. This is simpler mathematically than the conventional set of assumptions, since we no longer have to deal with the one-sided constraints of non-positive excess demand and non-negative prices. Instead, the (unconstrained) prices are such that the excess demand is zero, which is a simpler, and perhaps more realistic, version of the price equilibrium problem.

In von Neumann's original model, the only reference to capital goods concerns the replacement of worn-out capital, and this puts capital goods on the same footing as all other goods. In the present paper, however, we take into account that a process of production needs to acquire capital goods in order to increase the rate at which it is running, and can sell off capital goods when it is reducing the rate at which it is running. This couples the rate of acquisition of capital to the *rate of change* of the rate of production. This makes capital goods fundamentally different from other goods, since the need for an ordinary input to a production process is proportional to the rate at which the process is running, and not to the rate of change of that rate. Besides the foregoing, we *also* take into account the need to replace worn-out capital, but this appears in our model in a similar way to all of the non-capital inputs to a production process. In particular this need is proportional to the rate of production, and not to the rate of change of that rate.

By including capital goods in the manner described above, we make it possible for our model to run in continuous time, and to be described by differential equations. The original model of von Neumann is necessarily a discrete-time model, since there are no quantities in it with dimensions of time other than what might be called the production delay, i.e., the time between the acquisition of raw materials and the output of goods that can be brought to market. Thus, a dynamic version of von Neumann's original model necessarily alternates between production periods and what might be called market days, in which goods are exchanged and prices determined. In the present model, the need to acquire capital in order to change the rate of a process turns out to determine a time scale, and this ultimately sets the time scale of economic growth.

A limitation of the present model is that we consider only one capital good, which some processes produce and which all processes need in order to function. This simplification is made for mathematical reasons, but a possible excuse for making it is that the role of financial institutions, including especially the stock market, is what might be called the homogenization of capital, so that one form of capital can easily be exchanged for another, and all can be expressed in dollars. Thus, when a company wants to expand, we say that it needs to "raise

capital", and not that it needs to acquire specific machinery, buildings, etc. that are needed for the production process, although it is understood that the acquired capital (in dollars) will ultimately be used to purchase such equipment.

One of the benefits of having a model with only one capital good is that we can use the price of capital to set the scale of prices. Like many economic models, ours has the feature that only relative prices matter, and we normalize prices by setting the price of capital equal to one. This reflects the central role of capital in our model, since all processes need capital in order to function.

Although a discrete-time dynamical system is implicit in von Neumann's original model, he does not investigate its dynamics. What he does instead is to prove the existence of a *balanced-growth* solution, in which the economy expands at a fixed rate from one production period to the next. A distinctive feature of von Neumann's balanced-growth solution is that only some of the processes that define the economy are actually running, and only some of the goods have positive prices (the rest being free goods, since negative prices are not allowed). The uniqueness and stability of the balanced-growth solution are not investigated.

In the present paper, we have a continuous-time dynamical system, and we use numerical simulation to investigate its dynamics. At each timestep of the numerical method, there is a price-equilibrium problem to be solved, and we prove the existence and essential uniqueness of the solution under both versions of the model. (The prices may not be uniquely determined, but we prove that their non-uniqueness, if any, has no effect on the rate of change of the intensity of any process, so the development of the model economy is uniquely determined even if the prices themselves are not.) What we observe, but do not prove, is the evolution into a balanced-growth solution. During this evolution, prices gradually stabilize, many processes are gradually weeded out of of the evolving economy as the surviving processes become better and better tuned to each other, and the rate of economic growth gradually increases, eventually settling into a plateau in which the rate of economic growth is constant. This happens in both versions of the model, but with the following differences: In the first scenario, with prices constrained to be non-negative and with excess supply allowed, the final rate of economic growth is higher, but the economy is more monopolistic, with fewer processes holding a greater percentage of capital. In the second scenario, with perfect market clearing enforced and negative prices allowed, the plateau of economic growth is reached at a lower level, and the mature economy is more diverse. These qualitative results are very robust, in the sense that they persist despite different choices of the parameters that define the model economy.

The contributions of this paper are as follows. First, within the framework of the direction set by von Neumann, we have incorporated the distinctive nature of capital goods in setting the pace of economic growth, and we have thereby reformulated von Neumann's model in continuous time as a system of differential equations. We have used these equations to study the dynamical evolution of a model economy involving large numbers of processes and goods, in contrast to the small numbers that are often considered in economic theory, especially in textbooks [3–5]. In this evolution, we see a process of selection, in which many processes fail, while others succeed, and we consider it a contribution that we have been able to simulate this economic version of natural selection within a large and diverse model economy.

Another contribution of this paper is the manner in which prices are determined. The vague notion of "utility" is completely absent from our model, and yet prices are determined. Also, the computational methods that we use for the solution of the price-equilibrium problem are very straightforward in comparison to those that have been previously employed [6, 7]. To be fair, our methods are somewhat specific to the versions of the price-equilibrium problem that we consider, but the fact that we can determine the prices at every time step of a long-duration simulation involving a large number of processes and goods with just a few lines of

code and in a modest amount of computing time may also be counted as a contribution. Finally, with regard to prices, we have here introduced the notion of negative prices in relation to the removal of unwanted goods, and we have used this notion to study the impact of recycling laws on economic growth and diversity. To measure the effect on diversity, we have introduced a diversity index based on entropy in the distribution of capital, and this index may itself prove useful as an empirical tool.

These contributions are also significant in the real-world setting, and are particularly relevant to the current trends of commerce. Although the complete recycling scheme assumed by one of our price-equilibrium scenarios is not currently in use, nor likely to be implemented fully in the foreseeable future, nevertheless, the positive link that we have discovered between recycling and economic diversity is beginning to make itself felt. Indeed, the term *Upcycling* has been coined to describe the procedure of transforming otherwise discarded byproducts of production into valuable inputs, with market-determined prices, to the production of other goods. Upcycling businesses are fast-growing in recent years across multiple industries, and have indeed introduced diversity not only in the economic sense, but also in terms of providing new mindsets and tangible methods for the environmentally-conscious to approach their objective [8–10]. These current trends suggest that the link between recycling and economic diversity is real and significant, and they therefore lend support to the claim that our model has something useful to say about economic reality.

## Model set-up

We consider a model economy with $n$ processes indexed by $i = 1, \ldots, n$, and with $m + 1$ goods indexed by $j = 0, 1, \ldots, m$. The $0^{\text{th}}$ good is *capital equipment*, which is needed by all processes in order to run. It is, of course, a simplification to lump all kinds of capital equipment into a single category, but this has substantial mathematical benefits, which is why we make this simplification here.

The model economy runs in continuous time and is described by differential equations. This is only possible because we include the need for capital equipment in the model. The original von Neumann model of economic growth [1], see also [11, 12], does not have this feature and is therefore necessarily a discrete-time model.

Let $r_i(t) > 0$ be the rate at which process $i$ is running at time $t$. Also, let $p_j(t)$ be the price of good $j$ at time $t$. It is conventional in economic theory to assume that $p_j(t) \geq 0$, but the possibility of negative prices will also be considered here. The rationale for this will be discussed below, and two versions of the model will be developed in parallel, one with non-negative prices and the other in which the prices are unconstrained.

Our basic assumption is that the net rate of output of good $j$ by process $i$ is given by the following expression:

$$r_i(t)\pi_{ij} - \frac{dr_i}{dt}(t)c_i\delta_{j0} \tag{1}$$

for $i = 1, \ldots, n$ and $j = 0, \ldots, m$.

In (1), $\pi_{ij}$ and $c_i$ are given constants that characterize the model economy, with

$$c_i > 0, \quad i = 1, \ldots, n. \tag{2}$$

The constants $\pi_{ij}$ can be any real numbers. If $\pi_{ij} > 0$, then process $i$ is a net producer of good $j$; and if $\pi_{ij} < 0$, process $i$ is a net consumer of good $j$. In particular, if $\pi_{i0} > 0$, then process $i$ is a net producer of capital equipment. If $\pi_{i0} < 0$, then process $i$ is a net consumer of capital

equipment even when process $i$ is running at a constant rate. This represents the need to replace worn-out capital equipment.

The second term in (1) reflects our assumption that the amount of capital equipment needed by a process is proportional to the rate at which that process is running. Thus, in order to increase its rate, a process must acquire capital equipment. Conversely, when a process is decreasing its rate, it can sell off capital equipment in order to acquire revenue that can be used for the purchase of raw materials. Both of these effects give a kind of inertia to the rate at which a process is running, and thereby set the time scale of economic growth.

We assume that every process maintains a balanced budget at every time $t$. This gives the equation

$$\sum_{j=0}^{m} \left( r_i(t)\pi_{ij} - \frac{dr_i}{dt}(t)c_i\delta_{j0} \right)p_j(t) = 0, \quad i = 1, \ldots, n, \tag{3}$$

which can also be written as

$$c_i p_0(t)\frac{dr_i}{dt}(t) = r_i(t)\sum_{j=0}^{m}\pi_{ij}p_j(t), \quad i = 1, \ldots, n. \tag{4}$$

From this equation it is clear that only relative prices matter, so from now on we set

$$p_0(t) = 1, \tag{5}$$

and then (4) becomes

$$c_i\frac{dr_i}{dt} = r_i(t)\left( \pi_{i0} + \sum_{j=1}^{m}\pi_{ij}p_j(t) \right). \tag{6}$$

The *excess demand* for good $j$ at time $t$ is defined as the rate at which good $j$ is being consumed minus the rate at which good $j$ is being produced, that is,

$$e_j(t) = \delta_{j0}\sum_{i=1}^{n}\frac{dr_i}{dt}(t)c_i - \sum_{i=1}^{n}r_i(t)\pi_{ij}. \tag{7}$$

Since there is no storage of goods in our model, positive excess demand is impossible, and negative excess demand implies that some of what has been produced is simply thrown away without being consumed by any process.

The formula for $e_0(t)$ involves $\frac{dr_i}{dt}(t)$ and can therefore be simplified by making use of Eq (6). To do this, we replace $j$ by $k$ in (6) to avoid confusion with the $j$ in (7). Noting the cancelation of the terms involving $\pi_{i0}$, we get

$$e_0(t) = \sum_{i=1}^{n}r_i(t)\sum_{k=1}^{m}\pi_{ik}p_k(t). \tag{8}$$

Also, for $j > 0$, Eq (7) reduces to

$$e_j(t) = -\sum_{i=1}^{n}r_i(t)\pi_{ij}, \quad j = 1, \ldots, m. \tag{9}$$

If we multiply both sides of (9) by $p_j(t)$, sum over $j = 1, \ldots, m$, and add the results to the corresponding sides of Eq (8), we get

$$e_0(t) + \sum_{j=1}^{m} e_j(t)p_j(t) = 0. \tag{10}$$

This is known as *Walras' law* [13]. It states that the total monetary value of the excess demand is zero. Although we have not yet said anything about how the prices are determined, a very important remark is that Walras' law holds at *any* prices, not merely those that are determined by the conditions of price equilibrium. It is to these conditions that we turn our attention next.

We now state two versions of the conditions for price equilibrium. These will be denoted PE1 and PE2. The prices are constrained to be non-negative in PE1, and they are unconstrained in PE2.

PE1: $p_1(t), \ldots, p_m(t)$ are such that

$$p_j(t) \geq 0, \quad j = 1, \ldots, m, \quad t \geq 0, \tag{11}$$

$$e_j(t) \leq 0, \quad j = 0, \ldots, m, \quad t \geq 0. \tag{12}$$

Recall also that $p_0(t) = 1$, so all of the prices are non-negative under the conditions PE1.

When we combine the conditions PE1 with Walras' law, Eq (10), we get some interesting consequences. According to (11 and 12), every term on the left-hand side of (10) is non-positive, and such terms can only have zero as their sum if each of them is equal to zero. Therefore,

$$e_0(t) \quad = \quad 0, \quad t \geq 0, \tag{13}$$

$$e_j(t)p_j(t) \quad = \quad 0, \quad j = 1, \ldots, m, \quad t \geq 0. \tag{14}$$

From (14) we conclude that

$$e_j(t) < 0 \Rightarrow p_j(t) = 0, \quad j = 1, \ldots, m, \quad t \geq 0. \tag{15}$$

Thus, under PE1, any non-capital good for which there is negative excess demand (i.e., positive excess supply) becomes a *free good*. Note, however, that it is impossible to say before solving for the equilibrium prices, which goods will be free and which will have a positive price.

PE2: $p_1(t), \ldots, p_m(t)$ are such that

$$e_j(t) = 0, \quad j = 1, \ldots, m, \quad t \geq 0. \tag{16}$$

When we combine (16) with Walras' law, Eq (10), we see immediately that

$$e_0(t) = 0, \quad t \geq 0, \tag{17}$$

just as in PE1. Note that the prices are allowed to become negative under PE2, except for $p_0(t)$, which we have set equal to 1.

When $e_j(t) = 0$, we have *market clearing* for the good with index $j$. That is, the rate of production of good $j$ is exactly equal to the rate of consumption of good $j$. Under both of the conditions PE1 and PE2, we have market clearing for capital equipment. Under PE2, we have market clearing for all goods, but under PE1, we have only the weaker condition that the excess demands are non-positive for non-capital goods. As remarked above, non-positive excess demand is logically necessary, since there is no storage of goods in our model, so it is impossible to consume any good more rapidly than that good is being produced.

Under PE1, negative excess demand (i.e., positive excess supply) is allowed for non-capital goods. When this happens, it means that a good is being produced more rapidly than it is being consumed, with the price of that good falling to zero and the excess production of that good being thrown away.

As discussed in the Introduction, the overproduction and disposal of free goods may be harmful to the environment. Why would such goods be produced at all? In our model and also in reality, the production of a valuable good may be linked to the production of a good that has little use. If legislation is passed to prevent the disposal of the unwanted byproduct, then the byproduct may well develop a negative price. This, in turn, will encourage the growth of processes that can make use of the byproduct as a raw material, since the negative price of the byproduct makes it especially attractive. One specific example of the aforementioned procedure is the recycling and reuse of polyester. Plastic bottles are indeed harmful to the environment if they are allowed to be discarded freely after use. However, if the recycling of plastic bottles is enforced, they become a useful raw material for the clothing industry—now it isn't rare to find clothing brands that carry products made this way. More broadly speaking, the whole idea of *recycling* is aiming towards making useful material out of goods that would otherwise be unwanted, and become harmful to environment if allowed to be simply thrown away. This is the motivation for version PE2 of the price equilibrium problem, in which market clearing is enforced by legislative fiat, and in which prices are therefore allowed to be negative.

## Numerical methods

The mathematical formulation of our model economy is now complete (in two versions, PE1 and PE2), and we turn to the question of its numerical implementation. The simplest numerical scheme for Eq (6) is Euler's method:

$$c_i \frac{r_i(t + \Delta t) - r_i(t)}{\Delta t} = r_i(t) \left( \pi_{i0} + \sum_{j=1}^{m} \pi_{ij} p_j(t) \right), \qquad (18)$$

which can also be written as

$$r_i(t + \Delta t) = r_i(t) \left( 1 + \frac{\Delta t}{c_i} \left( \pi_{i0} + \sum_{j=1}^{m} \pi_{ij} p_j(t) \right) \right). \qquad (19)$$

We assume that $\Delta t$ is small enough that the following inequality is satisfied:

$$1 + \frac{\Delta t}{c_i} \left( \pi_{i0} + \sum_{j=1}^{m} \pi_{ij} p_j(t) \right) > 0, \qquad (20)$$

for $i = 1, \ldots, n$ and for all $t \geq 0$. The simplest way to enforce this is to monitor the $r_i(t)$, and to stop the computation and restart with a smaller $\Delta t$ if any one them ever becomes negative or zero. (One could also use an adaptive time step with (17) as a requirement that $\Delta t$ has to satisfy at each step.) With this restriction enforced, we may assume from now on that the $r_i(t)$ are positive for all $t$. Note, however, that some of them may approach zero as $t \to \infty$.

The discretization of Eq (7), which defines the excess demand for good $j$ at time $t$, is now as follows:

$$
\begin{aligned}
e_j(t) &= \delta_{j0}\sum_{i=1}^{n} c_i \frac{r_i(t+\Delta t) - r_i(t)}{\Delta t} - \sum_{i=1}^{n} r_i(t)\pi_{ij}, \\
&= \delta_{j0}\sum_{i=1}^{n} r_i(t)\left(\pi_{i0} + \sum_{k=1}^{m}\pi_{ik}p_k(t)\right) - \sum_{i=1}^{n} r_i(t)\pi_{ij}
\end{aligned}
\tag{21}
$$

in which we have used (18) to eliminate $\dfrac{r_i(t+\Delta t) - r_i(t)}{\Delta t}$. When $j = 0$, the second line of Eq (21) is the same as Eq (8); and when $j > 0$, it is the same as Eq (9). Thus, a nice consequence of using Euler's method is that we get the same formulae for the excess demands regardless of whether we derive those formulae before or after discretization.

Next, we need to discuss the discretization of the price-equilibrium conditions, PE1 and PE2. A peculiarity of Eq (9) is that it does not explicitly involve the prices. Thus, if the $r_i(t)$ are already known at time $t$, then the excess demands for non-capital goods are already determined, independent of the prices at time $t$. To remedy this, we evaluate the excess demands at time $t + \Delta t$ in terms of the prices at time $t$:

$$
\begin{aligned}
e_j(t+\Delta t) &= -\sum_{i=1}^{n} r_i(t+\Delta t)\pi_{ij} \\
&= -\sum_{i=1}^{n} r_i(t)\left(1 + \frac{\Delta t}{c_i}\left(\pi_{i0} + \sum_{k=1}^{m}\pi_{ik}p_k(t)\right)\right)\pi_{ij} \\
&= e_j(t) - \Delta t\left(b_j(t) + \sum_{k=1}^{m} A_{jk}(t)p_k(t)\right),
\end{aligned}
\tag{22}
$$

for $j = 1, \ldots, m$, where

$$
A_{jk}(t) = \sum_{i=1}^{n} \frac{r_i(t)}{c_i}\pi_{ij}\pi_{ik}, \quad j, k = 1, \ldots, m,
\tag{23}
$$

$$
b_j(t) = \sum_{i=1}^{n} \frac{r_i(t)}{c_i}\pi_{i0}\pi_{ij}, \quad j = 1, \ldots, m..
\tag{24}
$$

The $m \times m$ matrix $A(t)$ with elements $A_{jk}(t)$ is obviously symmetric, and it is also non-negative definite, since for any $q \in \mathbb{R}^m$ we have

$$
\begin{aligned}
q^{\mathrm{T}}A(t)q &= \sum_{j,k=1}^{m} q_j A_{jk}(t)q_k \\
&= \sum_{i=1}^{n} \frac{r_i(t)}{c_i}\left(\sum_{j=1}^{m}\pi_{ij}q_j\right)^2 \geq 0,
\end{aligned}
\tag{25}
$$

since $r_i(t)$ and $c_i$ are both positive for all $i$.

For any $q$ in the null space of $A(t)$, we obviously have equality in (25). Thus,

$$A(t)q = 0 \Rightarrow \sum_{j=1}^{m} \pi_{ij} q_j = 0 \text{ for } i = 1, \dots, n. \tag{26}$$

Also, from (23) it is obvious that

$$\sum_{j=1}^{m} \pi_{ij} q_j = 0 \text{ for } i = 1, \dots, n \Rightarrow A(t)q = 0. \tag{27}$$

Thus the null space of $A(t)$ is independent of $t$ and is given by

$$\text{null}(A(t)) = \left\{ q \in \mathbb{R}^m : \sum_{j=1}^{m} \pi_{ij} q_j = 0 \text{ for } i = 1, \dots, n \right\}. \tag{28}$$

Let

$$\phi(p, t) = \frac{1}{2} p^{\mathrm{T}} A(t) p + p^{\mathrm{T}} \left( b(t) - \frac{1}{\Delta t} e(t) \right). \tag{29}$$

Here and in the following, $p$, $b$, and $e$ denote vectors in $\mathbb{R}^m$ with components numbered from $1, \dots, m$. In particular, components with the subscript 0 are not included.

For each $t$, $\phi(p, t)$ is a convex function of $p$, since $A(t)$ is non-negative definite, but $\phi(p, t)$ may not be strictly convex, since $A(t)$ may have a non-trivial null space.

In terms of $\phi$, Eq (22) becomes

$$e_j(t + \Delta t) = -\Delta t \frac{\partial \phi}{\partial p_j}(p(t), t), \quad j = 1, \dots, m, \tag{30}$$

or in vector notation,

$$e(t + \Delta t) = -\Delta t (\nabla \phi)(p(t), t). \tag{31}$$

We are now ready to state the versions of PE1 and PE2 that will be used in our numerical scheme. These will be called PE1($\Delta t$) and PE2($\Delta t$). Both involve the minimization of $\phi(, t)$ at each time step, but the minimization is constrained in PE1($\Delta t$) and unconstrained in PE2($\Delta t$).

PE1($\Delta t$): At each time $t = 0, \Delta t, 2\Delta t, \dots$, choose $p(t) \in \mathbb{R}^m$ to minimize $\phi(p, t)$ subject to the constraint that $p(t) \geq 0$. (Here and in the following, an inequality involving a vector is understood to mean the corresponding inequality for each component.).

PE2($\Delta t$): At each time $t = 0, \Delta t, 2\Delta t, \dots$, choose $p(t) \in \mathbb{R}^m$ to minimize $\phi(p, t)$.

The problem PE1($\Delta t$) is a quadratic programming problem that can be solved by the Matlab function `quadprog`. [14] The problem PE2($\Delta t$) requires only the solution of a system of linear equations. The existence of a solution can be proved in each case, see Appendix 1, where it is also shown in both cases that any non-uniqueness is harmless, since different solutions $p(t)$ in the case that $p(t)$ is non-unique yield the same values of $r_i(t + \Delta t)$, so the non-uniqueness, if any, has no effect on the evolution of the model economy.

The Matlab code used for finding $p$ at every time step under both versions are provided here for convenience: (Note that `e`, `A` and `b` are assumed to have been constructed per (21), (23), and (24). `dt` represents $\Delta t$).

PE1($\Delta t$):

```
for j = 1:m  //note the exclusion of capital good
  f(j) = b(j)−e(j)/dt;
end
p = quadprog(A,f,[],[],[],[],zeros(m,1),[]);
```

PE2($\Delta t$):
```
for j = 1:m //note the exclusion of capital good
  f(j) = b(j)−e(j)/dt;
end
p = -A\f;
```

Next, we discuss the relationship between PE1($\Delta t$) and PE1, and then we will similarly discuss the relationship between PE2($\Delta t$) and PE2.

Under PE1($\Delta t$), we have $p(t) \geq 0$ by definition, and this is the same as in PE1. Also, since $p(t)$ minimizes $\phi$ over $p \geq 0$, we have

$$\frac{\partial \phi}{\partial p_j}(p(t)) \geq 0, \quad j = 1 \dots m \tag{32}$$

with

$$p_j(t)\frac{\partial \phi}{\partial x_j} = 0, \quad j = 1 \dots m \tag{33}$$

In terms of $e_j(t + \Delta t)$, these equations become

$$e_j(t + \Delta t) \leq 0, \quad j = 1 \dots m, \quad t = 0, \Delta t, 2\Delta t, \dots \tag{34}$$

$$p_j(t)e_j(t + \Delta t) = 0, \quad j = 1 \dots m, \quad t = 0, \Delta t, 2\Delta t, \dots \tag{35}$$

The inequality (34) is the same as the inequality (12) of PE1, except that here the time argument is $t + \Delta t$ instead of $t$, and also the values of $j$ in (34) are $1 \dots m$, whereas the inequality (12) of PE1 is also applicable to $j = 0$.

The shift of $t$ by $\Delta t$ in (34) in comparison to (12) makes no difference except that (34) does not include the condition that $e_j(0) \leq 0$. The setup of initial conditions will be discussed later.

The implications of (34 and 35) for $e_0$ are derived as follows. If we multiply both sides of (22) by $p_j(t)$, make use of (35). and sum over $j = 1 \dots m$, we get

$$0 = \sum_{j=1}^{m} e_j(t)p_j(t) - \Delta t \left( \sum_{j=1}^{m} b_j(t)p_j(t) + \sum_{j,k=1}^{m} A_{jk}(t)p_j(t)p_k(t) \right). \tag{36}$$

The first term on the right-hand side of this equation is equal to $-e_0(t)$ because of Walras' law, Eq (10), which holds for our numerical scheme just as it does for the original differential equations. Therefore, making use of the definitions of $b_j(t)$ and $A_{jk}(t)$, Eqs (23 and 24), we have

$$e_0(t) = -\Delta t \sum_{i=1}^{n} \frac{r_i(t)}{c_i} \left( \pi_{i0} \sum_{j=1}^{m} \pi_{ij}p_j(t) + \left( \sum_{j=1}^{m} \pi_{ij}p_j(t) \right)^2 \right). \tag{37}$$

Eq (37) shows that

$$e_0(t) = \mathcal{O}(\Delta t), \tag{38}$$

which is the discretized counterpart of the continuous equation $e_0(t) = 0$, Eq (13). It should also be noted that $e_0(t)$ plays no role in the dynamics of the discretized system; it is merely an output variable.

One might guess that $e_0(t)$ should never be positive, because of condition (12) of PE1, but actually

$$e_0(t) = -\sum_{j=1}^{m} e_j(t) p_j(t), \tag{39}$$

and since $e_j(t) \leq 0$ and $p_j(t) \geq 0$ for $j = 1\ldots m$, we reach the opposite conclusion that

$$e_0(t) \geq 0. \tag{40}$$

In the continuous case, we had $e_j(t)p_j(t) = 0$ and then (39) implied $e_0(t) = 0$, but here we only have $e_j(t)p_j(t - \Delta t) = 0$, see (39). This makes it clear precisely how the (positive) error in $e_0(t)$ can arise. Consider a time $t$ and a value of $j$ for which $e_j(t) < 0$ and therefore $p_j(t - \Delta t) = 0$. If $p_j(t) > 0$, then $-e_j(t)p_j(t) > 0$, and this makes a positive contribution to $e_0(t)$. Thus, $e_0(t) > 0$ in our numerical scheme if any good that was free at time $t - \Delta t$ is no longer free at time $t$. At most time steps, this will not be the case, and we will have $e_0(t) = 0$. When it is the case, although $e_0(t)$ will then be positive, it will be $\mathcal{O}(\Delta t)$.

In summary, we have the following comparison between PE1 and PE1($\Delta t$). In this comparison, $j \in \{1\ldots m\}$.

|  | PE1 | PE1($\Delta t$) |
|---|---|---|
| $p_j(t)$ | $\geq 0$ | $\geq 0$ |
| $p_0(t)$ | 1 | 1 |
| $e_j(t)$ | $\leq 0$ | $\leq 0$ |
| $e_0(t)$ | 0 | $\geq 0, \mathcal{O}(\Delta t)$ |
| $p_j(t)e_j(t)$ | 0 | $\leq 0, \mathcal{O}(\Delta t)$ |
| $p_j(t)e_j(t + \Delta t)$ | $\leq 0, \mathcal{O}(\Delta t)$ | 0 |

Now we turn our attention to the comparison of PE2 and PE2($\Delta t$). In PE2($\Delta t$) there is no constraint on the prices of non-capital goods, and that makes everything simpler. The minimization of $\phi(p, t)$ is accomplished by solving the symmetric non-negative definite linear system

$$A(t)p(t) = \frac{e(t)}{\Delta t} - b(t), \tag{41}$$

with $A(t)$ and $b(t)$ given by Eqs (23 and 24), and with $e(t) = e_1(t)\ldots e_m(t)$. It is easy to show, see Appendix 1, that the right-hand side of (41) is orthogonal to the null space of $A(t)$, so there exists at least one price vector $p(t)$ satisfying (41) at each time $t$. It is also shown in Appendix 1 that the non-uniqueness, if any, of $p(t)$ has no effect on the evolution of the model economy.

As a consequence of (41) we have

$$e_j(t + \Delta t) = 0 \ \text{ for } j = 1\ldots m, \ t = 0, \Delta t, 2\Delta t, \ldots, \tag{42}$$

see Eq (22). Thus, $e_j(t) = 0$ for $j = 1\ldots m$ for all time steps except possibly $t = 0$ (see below for a discussion of initial conditions) in PE2($\Delta t$) just as in the continuous formulation PE2. It follows from this and from Walras' law, Eq (10), that $e_0(t) = 0$, also. Since $e(t) = 0$, one may think that the term $e(t)/\Delta t$ may be omitted from the right-hand side of Eq (41). Indeed, this is true in principle, provided that $e(0) = 0$, since it then follows by induction that $e(t) = 0$ at every step. We think it is better, however, to retain the term $e(t)/\Delta t$ on the right-hand side of (41), since that makes the scheme self-correcting for any errors that may contaminate $e(t)$ and make it nonzero. In particular, the scheme as written will work even if $e(0)$ is nonzero, although it is better to avoid this since a large initial transient would thereby be introduced.

The comparison between PE2 and PE2($\Delta t$) may now be summarized as follows (with $j \in \{1 \ldots m\}$):

|  | PE2 | PE2($\Delta t$) |
| --- | --- | --- |
| $p_0(t)$ | 1 | 1 |
| $p_j(t)$ | unconstrained | unconstrained |
| $e_0(t)$ | 0 | 0 |
| $e_j(t)$ | 0 | 0 |

Thus, in PE2($\Delta t$) as in PE2, we have perfect market clearing, enforced by possibly negative prices.

## Construction of the model economy

The first step in constructing a model economy is to choose the parameters $n$ and $m$. We suggest

$$n >> m. \tag{43}$$

What we expect will then happen is a process of selection, in which $r_i(t) \rightarrow 0$ for many processes, but in which some processes are able to grow and form a viable economy. In the simulations reported here, $n = 1000$ and $m = 100$.

The next step is the construction of the matrix $\pi$. We let each element of $\pi$ be chosen as an independent Gaussian random variable with mean 0 and variance 1. This can easily be done in Matlab using the `randn` function. Fig 1 is a sample distribution of a column of $\pi$.

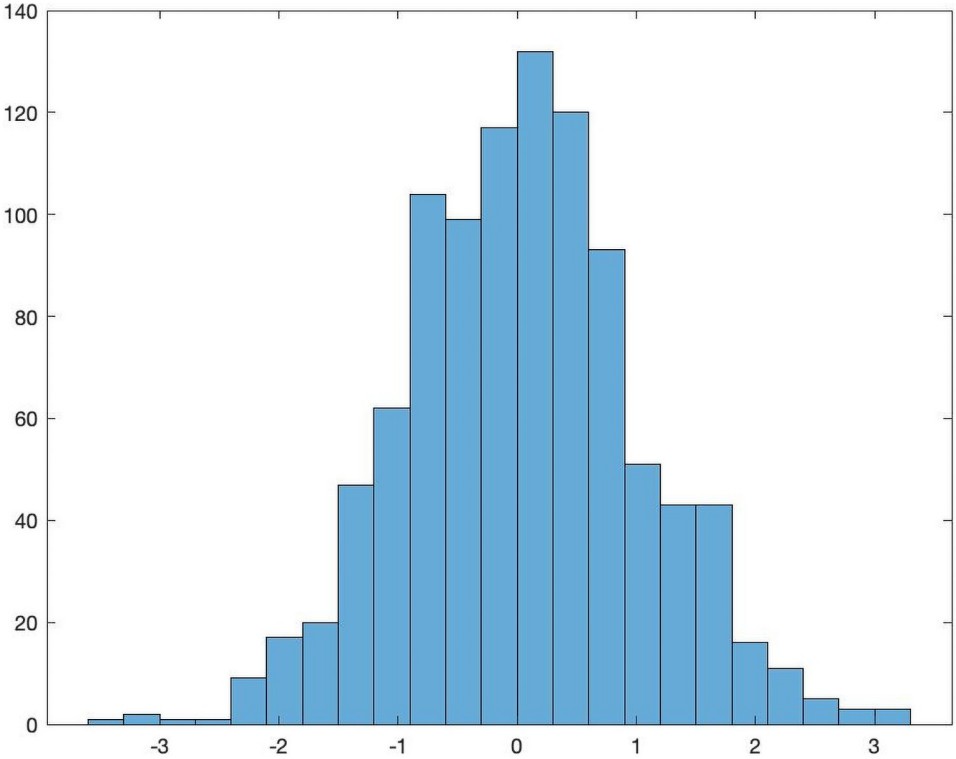

**Fig 1. Sample distribution of a column of the production matrix $\pi$.**

We also need to choose the coefficients $c_i > 0$ such that $c_i r_i$ is the amount of capital equipment that is needed by process $i$ when it is running at the rate $r_i$. We propose that each of the $c_i$ be chosen independently from some probability density function $\rho(c)$ that is supported on the positive real numbers. If all of the $c_i$ are scaled by some common factor, this results only in a change of time scale, see Eq (6), so there is no loss of generality in setting

$$\int_0^\infty c\rho(c)dc = 1. \tag{44}$$

Accordingly, we propose that every entry of $c$ to be chosen from a Gamma distribution with mean 1. With this restriction, we are still free to choose the variance of the Gamma distribution to set the extent to which processes in the model economy differ in their need for capital equipment. For this study, we choose a variance of 1/3. The Matlab command that generates the $c$ coefficients for all n processes is then

$$\mathtt{c \;=\; gamrnd(3, (1/3), [n, 1]); ,} \tag{45}$$

Fig 2 is a sample distribution of $c$ with mean of 1 and variance of 1/3.

The last step in constructing the model economy is to set-up the initial $r$ vector. Although it is not absolutely necessary to do so, we think it best to initialize the economy in a state of perfect market clearing. Because of Walras' law, this can be achieved by making

$$e_j(0) = 0, \quad j = 1, \ldots, m, \tag{46}$$

and then $e_0(0) = 0$ will follow.

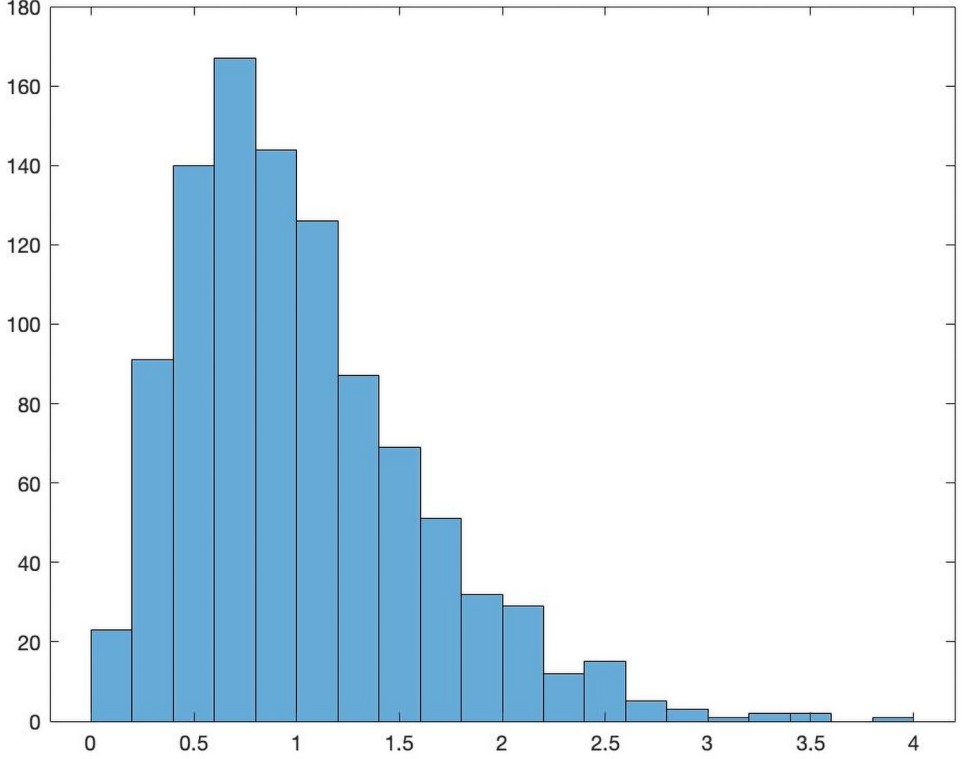

**Fig 2. Sample distribution of the capital requirement coefficients $c$.**

From Eq (9), we can see that (46) is equivalent to

$$\sum_{i=1}^{n} r_i(0)\pi_{ij} = 0, \;\; j = 1, \ldots, m. \tag{47}$$

Thus $r(0)$ must lie in the left null space of the $n \times m$ submatrix that is obtained from $\pi$ by deleting column 0, which is the column that refers to capital goods. Since we also need $r_i(0) \geq 0$ for $i = 1, \ldots, n$, the requirement is not only that this submatrix must have a non-trivial left null space, but also that this null space must contain at least one vector with components that are all non-negative.

To find an eligible $r$ vector, we observe that the construction of the $\pi$ matrix results in each column having a sum whose expected value is zero. Consequently, Eq (53) is approximately satisfied when all components of $r(0)$ are equal. Therefore, for each particular choice of $\pi$, we look for an $r(0)$ vector that is closest in Euclidean norm to the vector of length $n$ that has all of its components equal to 1, while imposing the constraints that Eq (53) should hold and that all components of $r$ should be non-negative. We can perform this construction in Matlab using `quadprog` with the following syntax.

```
pi_new = pi(:,1:100); %pi deleting capital column
u = ones(1,n);
H = eye(n);
f = zeros(1,n);
Aeq = pi_new';
Beq = -pi_new'*u';
lb = (zeros(1,n)-u)';
w = quadprog(H,f,[],[],Aeq,Beq,lb,[]);
r = w'+u;
```

Note that a process $i$ for which $r_i(0) = 0$ is one that is not running at $t = 0$, and in our model, such a process will have $r_i(t) = 0$ for all $t > 0$, so we can consider such a process as if it did not exist at all. For this reason, it is important that our procedure for setting the values of $r_i(0)$ should not make too many of them equal to zero. To verify that not too many processes are being turned off at the start by this procedure, 5,000 random choices of the $\pi$ matrix were made, and a corresponding $r$ vector was constructed for each of them in the manner described above. In the worst case, 12/1000 components of $r(0)$ were set equal to zero, so we conclude that this starting procedure does not arbitrarily eliminate too many processes. Fig 3 is a sample distribution of $r(0)$.

## Results

We have conducted 30 pairs of simulations. Within each pair, the production matrix $\pi$, the capital requirement coefficients $c$, and the initial process intensities $r(0)$ are identical, having been chosen randomly by the procedures outlined above, but the same randomly constructed economy and the same initial condition is used for both members of each pair. The two members of the pair differ only in that prices are determined by PE1($\Delta t$) in the first case and by PE2($\Delta t$) in the second. Thus, for the first member of each pair, prices are nonnegative and excess demand is nonpositive; and for the second member of each pair, prices are unconstrained and market clearing prevails so that there is neither excess demand nor excess supply. The differences from one pair to another reflect only the randomness in the procedure that we have used to construct the model economy.

Results will be presented in terms of the following functions of time: process intensities, prices, economic growth, and an index of diversity that will be defined below. Note that time is

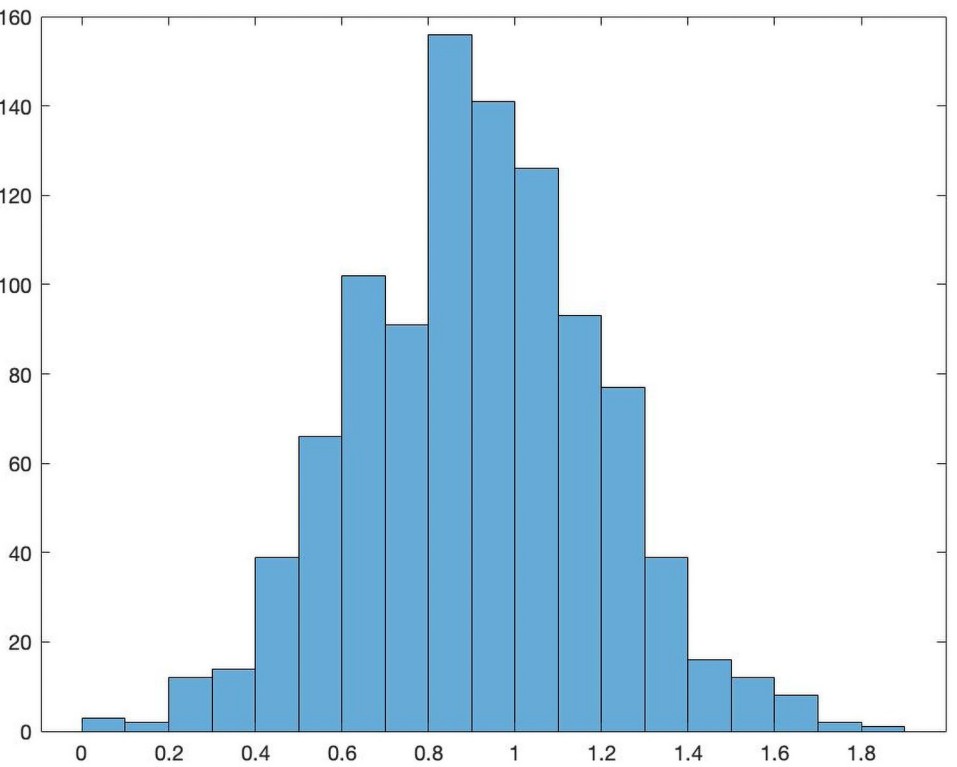

**Fig 3. Sample distribution of the initial process intensities $r(0)$.**

in arbitrary dimensionless units. In some cases we show typical results from one or more pairs of simulations, and in others we show results that summarize what happened in all 30 of the simulated pairs. A convergence test has been conducted on one of the 30 pairs of simulations, see Appendix 2.

## Process intensities

The graphs in Fig 4 are obtained by plotting the log of the intensity vector of processes ($r$ value) against time. It can be seen that a few of the processes have very small intensities (very negative $\log(r)$ values). These are processes that were essentially turned off by the initialization procedure described above, but because of the use of a finite tolerance by `quadprog`, their

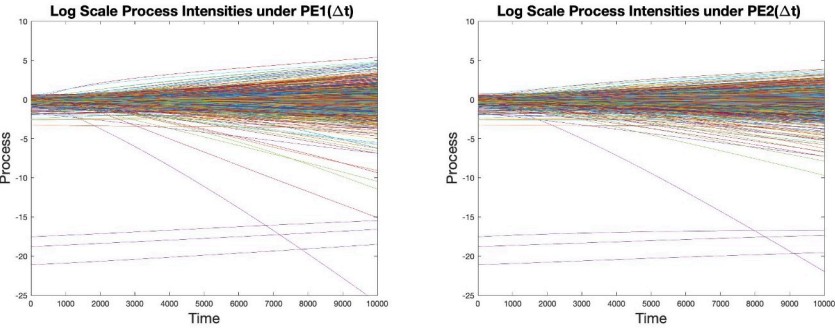

**Fig 4. Log scale process intensities under both PE scenarios.**

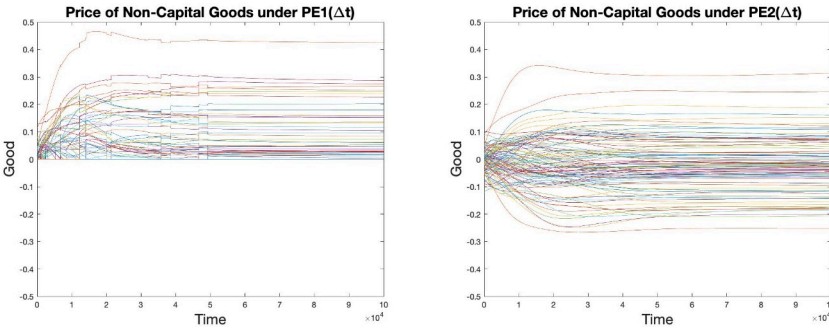

**Fig 5. Prices of non-capital goods under both PE scenarios.**

intensities were actually set equal to very small positive values. These processes have essentially no effect on the rest of the model economy.

Note that a straight line on the plot of $\log(r)$ as a function of $t$ indicates exponential growth, and the slope of the line is the growth rate. When two or more processes have straight lines with the same slope, that indicates that they are growing at the same rate. This is the *balanced-growth* state predicted by von Neumann, and it seems to occur for a subset of the processes in our model economies, i.e., for those with the highest growth rates. What we do not see, however, is a clean separation of process into two groups, with one group having a common maximal growth rate and with the rest of the processes falling behind. Instead, there seems to be a smooth distribution of growth rates. This is true under both price-equilibrium scenarios.

## Prices

The graphs in Fig 5 are obtained by plotting the price vector of goods against time. It can be seen that in PE1($\Delta t$), prices go as low as zero, representing free goods with positive excess supply, whereas in PE2($\Delta t$), prices of some goods do indeed go to negative, corresponding to our construction of enforced market clearing. A series of sudden movements of various magnitudes shared by all goods can be observed in PE1 at earlier stages of the simulation. These "jumps" in prices correspond with occasions in the economy where the price of a previously free good turns positive.

Graphs in Fig 6 are obtained by plotting the set of prices of non-capital goods under PE2($\Delta t$) against the set of prices of these goods under PE1($\Delta t$) at the end of each pair of simulations. 6 out of 30 pairs of simulations have been randomly selected and their data plotted here. In each plot, a good is represented as a dot with the x-coordinate denoting its price under PE1($\Delta t$), and the y-coordinate its prices under PE2($\Delta t$). It is clear that in each trial, there is some correlation between the two sets of prices, but also a lot of scatter. More interestingly, note that free goods under PE1($\Delta t$) can have positive as well as negative prices under PE2($\Delta t$)—these are the dots on the y-axis. Conversely, goods with negative prices under PE2($\Delta t$) are not necessarily free goods under PE1($\Delta t$)—these goods correspond to the dots under the line $y = 0$ and not on the y axis. The significant rearrangements of prices between the two price-equilibrium schemes of an otherwise identical economy demonstrate the complexity of the models as they evolve through time.

## Economic growth

Let $C(t)$ be the total amount of capital in the economy at time $t$:

$$C(t) = \sum_{i=1}^{n} c_i r_i(t). \tag{48}$$

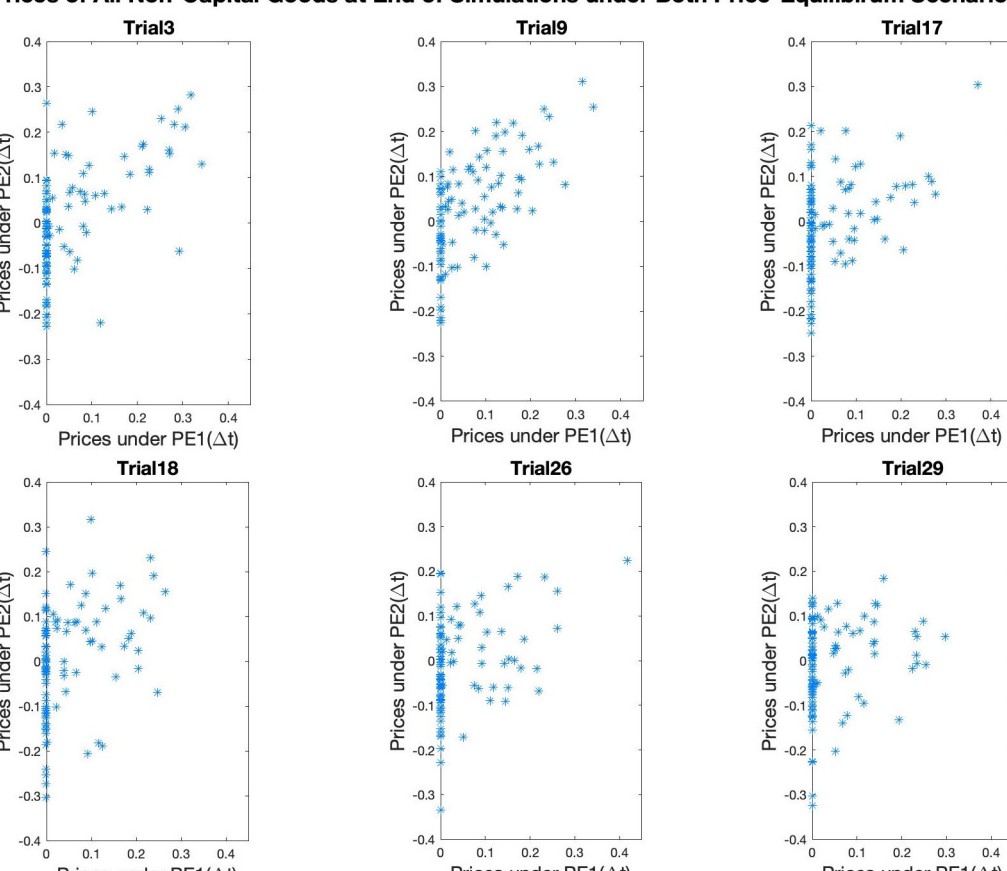

**Fig 6. Prices at end of simulations under both PE scenarios from randomly selected trials.**

Then the relative growth rate of the whole economy is defined by

$$G(t) = \frac{1}{C}\frac{dC}{dt} \tag{49}$$

Since the time-scale of our model is arbitrary, no absolute meaning can be attached to the number $G(t)$, but the two model economies of each pair can be compared by comparing their growth rates. This is done in Fig 7.

It can be seen that both models reach Balanced-Growth, characterized by a constant growth rate, and PEl($\Delta t$) enjoys a higher growth rate than PE2($\Delta t$).

In Fig 8, the relative growth rates of economy in all thirty pairs of simulations are graphed, with orange curves representing those under PE1($\Delta t$), and blue curves representing those under PE2($\Delta t$). Although it is hard to identify each specific pair of simulations in this graph, the general performance of economy under PE1($\Delta t$) achieving a higher relative growth rate than it does under PE2($\Delta t$) is evident.

## Diversity

At any time $t$, the fraction of capital held by process $i$ is given by

$$\alpha_i(t) = c_i r_i(t)/C(t), \tag{50}$$

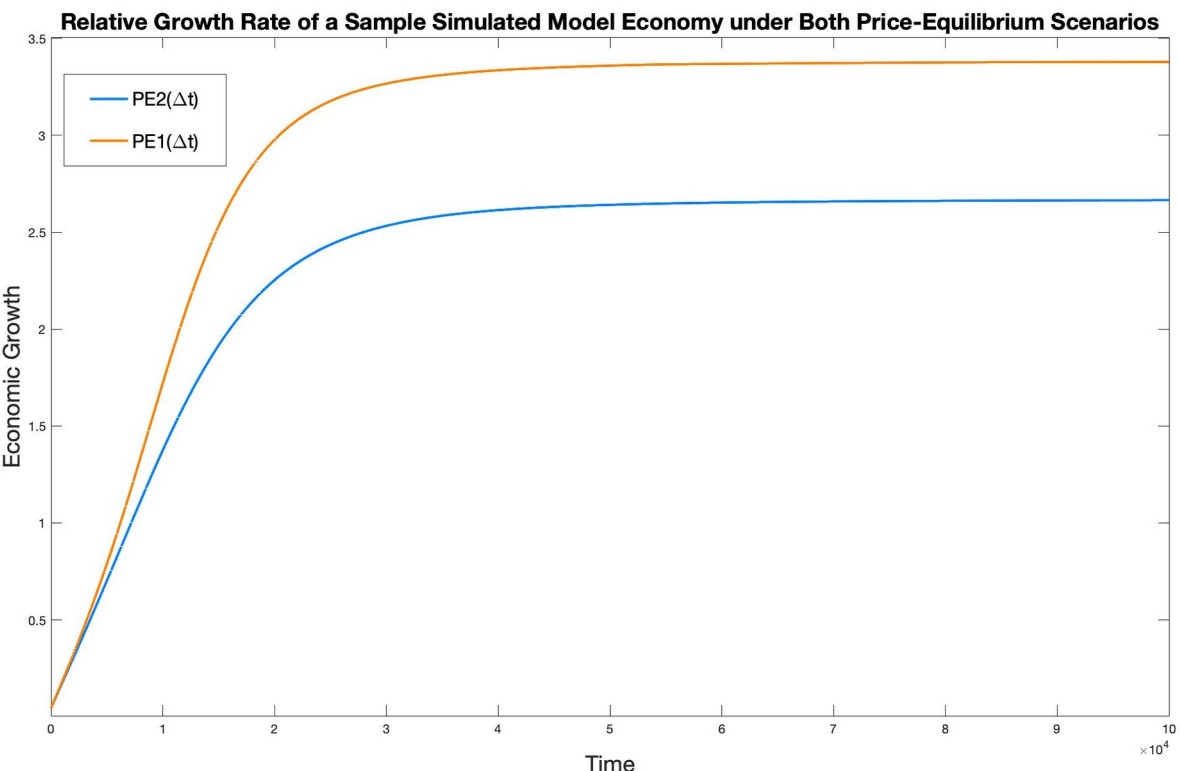

**Fig 7. Relative growth rates of a sample model economy under both PE scenarios.**

where $C(t)$ is the total amount of capital defined above by Eq (48). These fractions are plotted as functions of time in Fig 9.

Note that the fractions $\alpha_i(t)$ are nonnegative and also that their sum is 1, so in this respect they are like probabilities. This observation suggests that we define the *entropy* of the economy by

$$S(t) = -\sum_{i=1}^{n} \alpha_i(t) \log \alpha_i(t) \tag{51}$$

The entropy is zero when one process holds all of the capital in the economy, and it is maximized when all processes hold equal amounts of capital. The maximum possible value of the entropy with $n$ processes is

$$S_{\max} = -\sum_{i=1}^{n} \frac{1}{n} \log \frac{1}{n} = \log n \tag{52}$$

We therefore define the *diversity index*, denoted $D(t)$ by

$$D(t) = \frac{S(t)}{S_{\max}} \tag{53}$$

In this definition, the base that is used for the logarithm does not matter, and the possible values of $D(t)$ are in the interval $[0, 1]$.

Fig 10 is the diversity index of the two models plotted against time in one sample simulation.

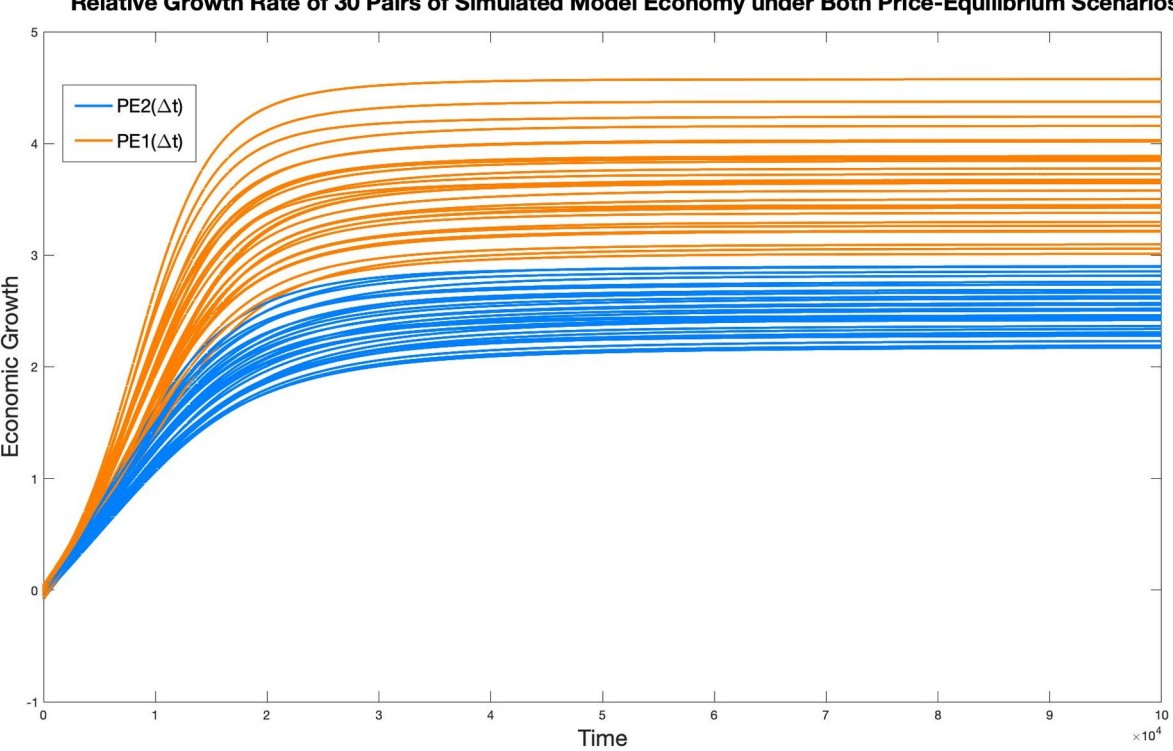

**Fig 8. Relative growth rates of all 30 simulated model economy under both PE scenarios.**

In Fig 11, the Diversity index of economy in all thirty pairs of simulations are graphed, with orange representing PE1($\Delta t$), and blue representing PE2($\Delta t$). Similar to the case in Fig 8, it is hard to identify each specific pair of simulations, however, the general performance of PE2($\Delta t$) achieving a more diverse economy than PE1($\Delta t$) is evident.

A distribution of the 30 pairs of model economy's Diversity Indices at the end of their respective runs is shown in the histogram in Fig 12. As discussed above, a model economy reaches a constant growth rate as it matures, therefore, these values, taken at the end of each simulation, are the Diversity Indices of each model economy at its Balance-Growth under both price-equilibrium scenarios. (Note that the scale for the Diversity Index values on the x-axis have been zoomed in so that the lack of any overlapping is evident).

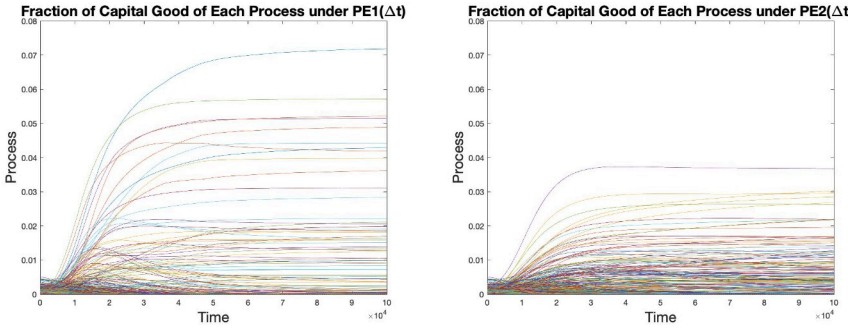

**Fig 9. Fraction of capital held by each process under both PE scenarios.**

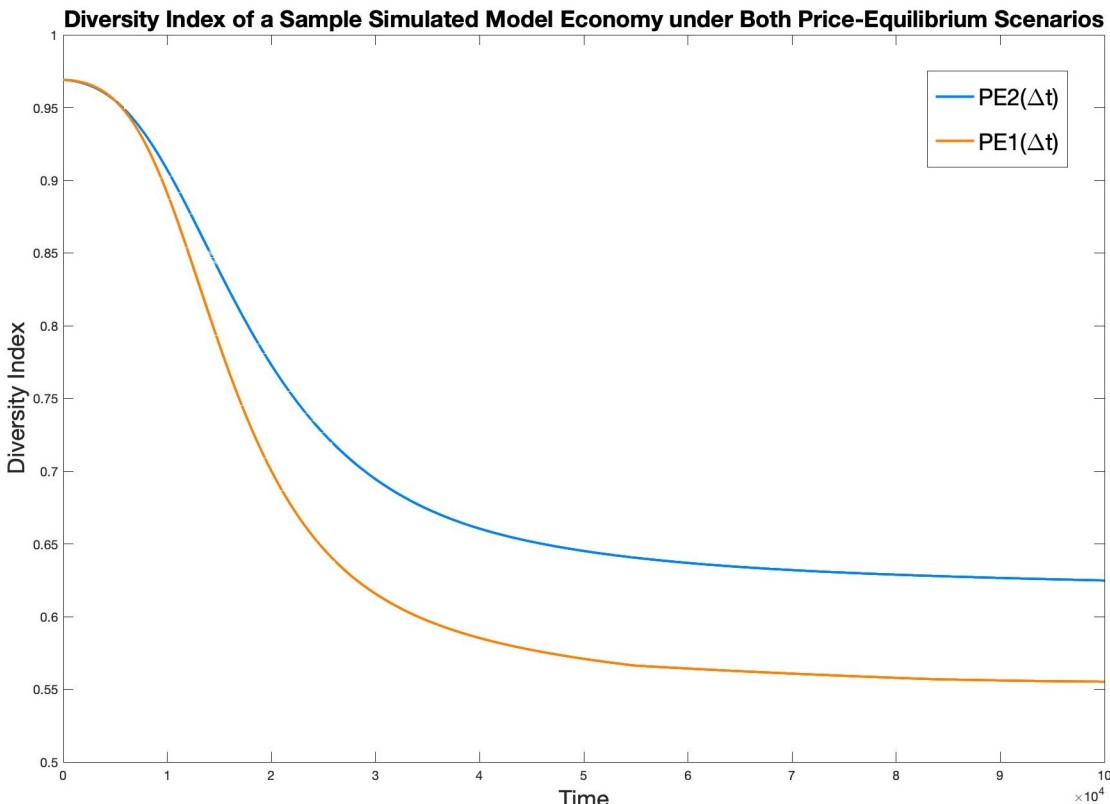

**Fig 10. Diversity index of a sample model economy under both PE scenarios.**

In the thirty sets of simulations we have run, the diversity index of PE1($\Delta t$), at balanced-growth, has a mean of 0.5523 and a standard deviation of 0.0274, whereas the diversity index of PE2($\Delta t$), at balanced-growth, has a mean of 0.6333 and a standard deviation of 0.0113.

## Discussion

Building a continuous-time model by adding a capital component to von Neumann's construction, and using this model to investigate the impact of universal recycling on economic growth and diversity, we provide a new perspective on three existing branches of economic studies: modern growth theories based on neoclassical concepts; extensions of von Neumann's model; and research on the economics of recycling.

Largely inheriting von Neumann's viewpoint that producers and consumers can be represented with the same construction of processes [1], our model provides a very different perspective on the macroeconomy from those based on the prevalent neoclassical foundation. This distinction has made it possible for us to consider a multitude of differentiated processes and goods, and their evolution as the market matures. In comparison to some neoclassical models, e.g. the Solow-Swan model [15, 16], the Ramsey-Cass-Koopmans model [17–19] and the Diamond model [20], whose discussions of the economy largely rely on an unrealistic uniformity among processes and goods, our model makes no such simplifying assumption. In fact, the support of large-scale numerical simulation allows us to welcome a high degree of differentiation.

Another noteworthy characteristic of our model, which could be considered a strength or a weakness, is the absence of any decision-making mechanisms. Such decision-making in the

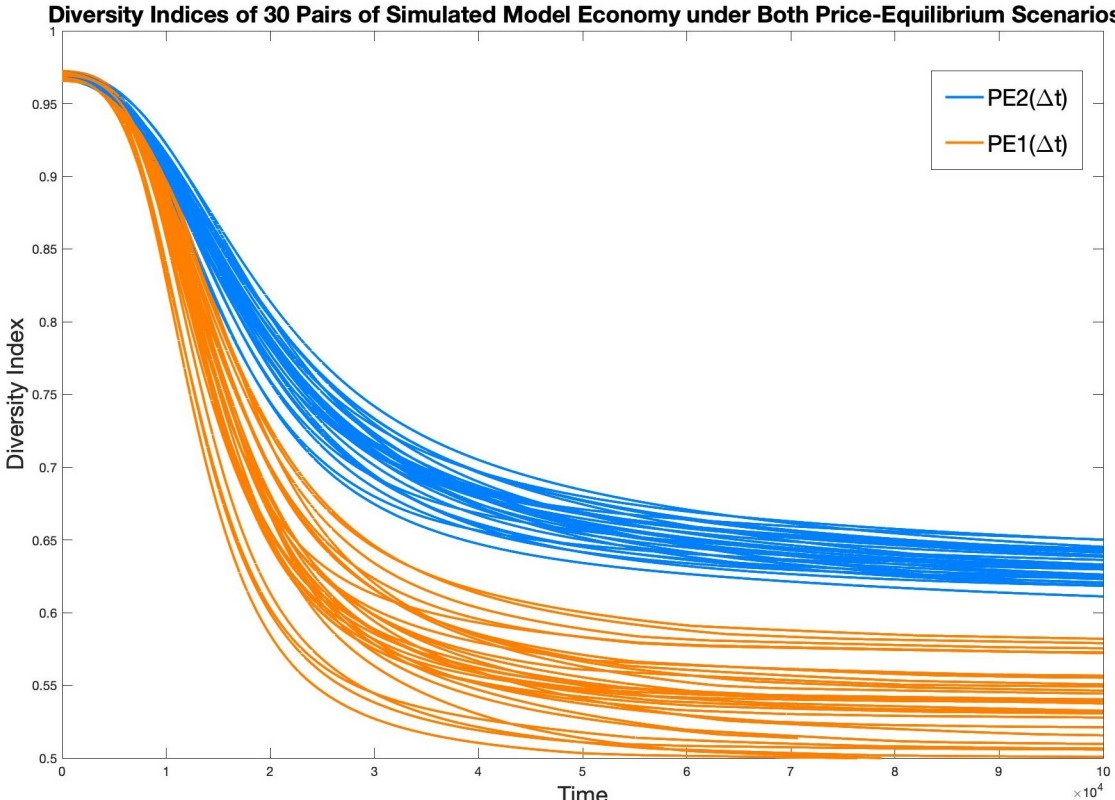

**Fig 11. Diversity indices of all 30 simulated model economy under both PE scenarios.**

modern growth theories, reflected by the optimization problems that guide the aggregate firm's behavior, see [15, 16, 21], and the aggregate households' behavior, see [17–20], is based on a fundamental assumption of neoclassical concepts that "individuals maximize utility and firms maximize profits" [22]. In our model, each process' production intensity is solely dependent on the budget constraint, as no storage is allowed. In other words, as some form of a budget constraint usually sets the upper limits of the intensity of the activity of a household or of a firm in neoclassical models, it is both the upper and lower limits in our model, and therefore dictates process intensities. In particular, our model has no a priori definition of utility, which is an integral part of neoclassical theory. Indeed, what might be called utility in our model is the extent to which a good may be useful in production, and this is economically determined, rather than exogenously given. It depends not upon characteristics of the good itself, but instead upon its relationship to other goods and processes in the economy as a whole. We believe this is a more realistic view of utility than the frankly naive concept that there is such a thing as the intrinsic value of a good, independent of economic relationships. Whatever one may think of the philosophical issues here, it is at least *interesting* that prices can be determined and economic growth simulated in a model that has no notion whatsoever of the intrinsic value of a good.

A weakness of our model in comparison to modern growth theories is the lack of flexibility given to processes in both production inputs and outputs. Contrary to neoclassical models which often assume "some positive and smooth elasticity of substitution between the inputs" [3], ours dictates fixed proportions of inputs. This is evidently far from reality as examples of substitutions among inputs are common: rubber and artificial rubber in the production of

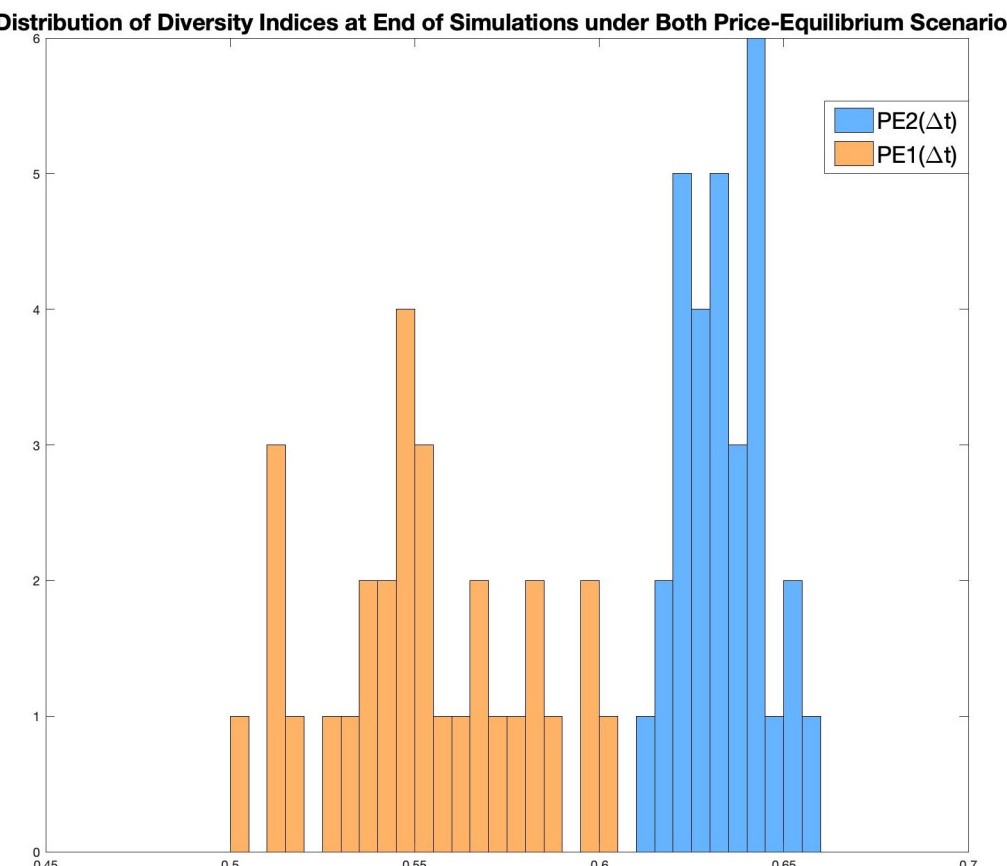

**Fig 12. Distributions of the diversity indices at the end of simulations for all 30 model economy under both PE scenarios.**

cars; butter and margarine in the production of sandwiches etc. Our model also suffers from the same rigidity on the output side. A process in our model cannot substitute the production of trucks for the production of cars, nor that of computers for that of cell phones.

Several extensions have been proposed to the original von Neumann's model, including efforts to build flexibility into the input and output matrices, see [23], and the construction of a stochastic version that is in other ways comparable to von Neumann's original model, see [24]. To our knowledge, our model is the first to suggest a new direction of extension in which the need to acquire capital equipment sets the pace of economic growth.

Significant efforts have been devoted to studying the economic impact of recycling. A relatively new recycling policy called *unit pricing*, linking waste disposal fees to some measurements of the waste—usually its weight or volume, is currently replacing the traditional flat-fee charged by waste disposal companies in various geographic locations [25]. The unit pricing system has the features that the amount of money that changes hands is proportional to the amount of goods that changes hands; money and goods are transferred in the same direction; and prices are determined by market forces. These attributes of the unit pricing system makes it an example of the price equilibrium scenario in our model where prices are allowed to be negative. Many existing works in this area of study focus on the analysis of empirical data to facilitate discussions on how better to implement recycling. Cases studies can be found that perform similar analysis on data from different locations and times [25, 26]. To our

knowledge, there has been no research, and in particular no mathematical models, that study the effect of enforced recycling on economic diversity.

A new business phenomenon, called Upcycling, converts unwanted byproducts that would otherwise be discarded into useful, market-priced inputs to the production of other goods. Upcycling is flourishing in multiple industries, especially the manufacturing of food, beverage, and clothing [8, 9]. One specific example of Upcycling is the use of the spent grain from brewing as flour for making bread. It has been common practice for breweries to give the spent grain to nearby farms as animal feed for free [10]. The introduction of Upcycling gives birth to new consumer- or business-facing companies specializing in converting the spent grain into flour. The end products often sell at a premium in comparison to other kinds of flour [9]. This example is closely related to our model in several ways.

The spent grain is an example of a free good under our first price-equilibrium scenario, but it can acquire a positive price if a process emerges that can make use of it. This illustrates a very important property of our model: that the goods that are free under the first (traditional) price-equilibrium scenario can sometimes acquire a positive price within the framework of universal recycling. Recall Fig 6, in which there are many examples of this phenomenon.

The most striking feature of Upcycling, as it is described in the cited news reports, is its contribution to economic diversity. Processes emerge that did not exist previously in order to make use of previously unwanted goods. This is precisely what our model predicts will happen.

Although the term "diversity" as used in this paper has a narrow technical meaning related to the distribution of capital among processes, the reports on Upcycling also make it clear that the new industries created in this way contribute to the diversity of economic opportunity, i.e., that these new industries make economic opportunity available to people who might have been excluded from more traditional businesses. Thus, it would not be wrong to interpret "diversity" more broadly and to say that a predicted consequence of universal recycling is a society in which economic opportunity is more universally shared.

## Summary & conclusion

In this paper, we have introduced and studied a model economy similar to the one originally proposed by von Neumann [1], in that the model considers *processes* and *goods*, without regard to the distinction between consumers and producers. We have generalized von Neumann's model by assuming that the rate at which a process is running determines its need for capital equipment, so that any increase in rate requires that capital equipment be purchased, and any decrease in rate allows for capital equipment to be sold in order to raise revenue. In this way we arrive at a model that can be formulated in continuous time, as a system of differential equations.

We have used the model described above as a framework for the study of two versions of the price equilibrium problem, denoted PE1 and PE2. PE1 makes the conventional assumptions, also made by von Neumann, that excess demand cannot be positive and that prices cannot be negative. PE2 assumes perfect market clearing, and to achieve this, leaves prices unconstrained, so that the price of a good may become negative under PE2.

Our model economies are described by a production matrix, a capital requirement vector, and vector of the rates at which all of the model processes are running at $t = 0$. These parameters that define the model economy are chosen randomly, from specified distributions. With these parameters fixed, we simulate the evolution of the *same* model economy under the two different price equilibrium scenarios PE1 and PE2. This comparison has been made for 30 different, randomly chosen economies. Under both price equilibrium scenarios, we have

observed the model economies maturing into a state of balance growth after a period of natural selection in which some processes are weeded out while others prosper. There are systematic differences, however, between the observed evolutions of the model economies under the two price equilibrium scenarios. In particular, we find that enforced market clearing with prices allowed to be negative slows economic growth but enhances the diversity of the economy. To measure diversity, we have introduced a *diversity index* based on the entropy of the distribution of the fractions of capital held by the different processes. What we find is that the entropy of the model economy systematically *decreases* as the economy self-organizes, and that this decrease is greater when goods are allowed to discarded than when universal recycling is enforced. The economic meaning of this is that universal recycling promotes economic diversity, presumably by creating a greater variety of opportunities for processes to prosper than would otherwise exist. As mentioned previously, one can consider the industry of making clothing out of synthetic fibers made from recylced bottles as an example of this phenomenon.

Our model is highly idealized, and has so many limitations that it would be impossible to discuss them all. Perhaps the most striking are the absence of any mechanism for the storage of goods and the absence of any financial institutions. Introduction of these features into the model will require the consideration of decision-making mechanisms that are currently not needed. For example, if a process does not need to bring all of its output to market immediately, it can decide how much to bring to market and how much to keep in storage to be brought to market at some later time. Similarly, a process might decide to buy more raw materials than it needs at the moment, and to store the excess for future use. The same idea is applicable to saving and borrowing. Instead of spending all of its revenue immediately, a process my decide to bank some of it for future use, and conversely, a process may decide to borrow money in order to run at a higher rate than would otherwise be possible based on the current revenue of that process.

In the present model, each process has fixed entries in the production matrix, and a fixed capital requirement for running at a given rate. Thus there is only one parameter for each process that can be adjusted as a function of time, namely the overall rate at which the process is running. A more realistic model would allow for some flexibility, so that an automobile manufacturer, for example, could decide what mixture of cars and trucks to produce, and change that mix based on projected market conditions. On the input side, the recipe of raw material for a given process may not be fixed, since one raw material can sometimes be substituted for another, and then decisions need to be made based on market conditions which raw materials to purchase.

It seems, then, that there is a long road ahead in terms of making the model more realistic. Nevertheless, the results reported here already show the value of having a self-contained economic model, one that grows and organizes itself without being told what to do, and on which different scenarios can be tried by computer experimentation.

## Appendix 1

In this appendix, we consider questions of existence and uniqueness related to the discretized price equilibrium problems PE1($\Delta t$) and PE2($\Delta t$). Recall that these problems involve the constrained and unconstrained minimization of the function $\phi(p, t)$ defined by Eq (29). Here, however, we drop the argument $t$, since we are dealing with a separate minimization problem at each of the times $t = 0, \Delta t, 2\Delta t, \ldots$. These problems differ only because of the different values of $r_i(t)$, which will here be denoted $r_i$. Thus, the objective function that we consider in this

appendix is

$$\phi(p) = \frac{1}{2}p^{\mathrm{T}}Ap + p^{\mathrm{T}}f, \tag{54}$$

where $f, p \in \mathbb{R}^m$ and $A$ is $m \times m$. The formulae for $A$ and $f$ are

$$A_{jk} = \sum_{i=1}^{n}\frac{r_i}{c_i}\pi_{ij}\pi_{ik}, \quad j, k = 1, \ldots, m, \tag{55}$$

$$
\begin{aligned}
f_j &= b_j - \frac{1}{\Delta t}e_j \\
&= \sum_{i=1}^{n}\frac{r_i}{c_i}\left(\pi_{i0} + \frac{c_i}{\Delta t}\right)\pi_{ij}, \quad j = 1, \ldots, m,
\end{aligned}
\tag{56}
$$

see Eqs (23 and 24) and (9).

Substituting (55) and (56) into (54), we get

$$\phi(p) = \sum_{i=1}^{n}\frac{r_i}{c_i}\left(\frac{1}{2}\left(\sum_{j=1}^{m}\pi_{ij}p_j\right)^2 + \left(\pi_{i0} + \frac{c_i}{\Delta t}\right)\sum_{j=1}^{m}\pi_{ij}p_j\right). \tag{57}$$

We claim that $\phi(p)$ is bounded from below. To show this, and obtain the lower bound, note that $r_i/c_i > 0$ for all $i$, and minimize each term by thinking of it as a function of $\sum_{j=1}^{m}\pi_{ij}p_j$. The minimum is achieved when

$$\sum_{j=1}^{m}\pi_{ij}p_j = -\left(\pi_{i0} + \frac{c_i}{\Delta t}\right), \tag{58}$$

and this gives the lower bound

$$\phi(p) \geq -\frac{1}{2}\sum_{i=1}^{n}\frac{r_i}{c_i}\left(\pi_{i0} + \frac{c_i}{\Delta t}\right)^2. \tag{59}$$

Note that (59) holds for all $p \in \mathbb{R}^m$.

It has already been noted in the main text that $A$ is postive semi-definite, with $Aq = 0$ if and only if $\sum_{j=1}^{m}\pi_{ij}q_j = 0$ for $i = 1, \ldots, n$. There may or may not be nontrivial $q \in \mathbb{R}^m$ that satisfy this condition, but we shall allow for the possibility that there are such $q$. The vector $f$ is orthogonal to the null space of $A$, since

$$\sum_{j=1}^{m}\pi_{ij}q_j = 0 \Rightarrow \sum_{j=1}^{m}f_jq_j = 0, \tag{60}$$

see (56).

Since $A$ is positive semi-definite, the function $\phi$ is convex, but it may not be strictly convex, since $A$ may have a nontrivial null space.

Let $\mathbb{R}^m_+$ denote the non-negative cone in $\mathbb{R}^m$:

$$\mathbb{R}^m_+ = \{p \in \mathbb{R}^m : p_j \geq 0 \text{ for } j = 1, \ldots, m\}. \tag{61}$$

We claim that

1. $\exists p^\star \in \mathbb{R}^m_+$ such that $\phi(p^\star) \leq \phi(p)$ for all $p \in \mathbb{R}^m_+$, with equality if and only if $p - p^\star$ is in the null space of $A$.

2. $\exists p^{\star\star} \in \mathbb{R}^m$ such that $\phi(p^{\star\star}) \leq \phi(p)$ for all $p \in \mathbb{R}^m$, with equality if and only if $p - p^{\star\star}$ is in the null space of $A$.

The only difference between the two claims is that in claim 1 we consider only $p \in \mathbb{R}_+^m$, but in claim 2 we consider all $p \in \mathbb{R}^m$.

The existence part of claim 1 is a special case of the Frank-Wolfe theorem [27]. We follow their method of proof here, but take advantage of the simple geometry of the domain $\mathbb{R}_+^m$. Another special feature of our situation is that the objective function $\phi$ is bounded from below over all of $\mathbb{R}^m$, not merely over $\mathbb{R}_+^m$. In the following, we exploit this to obtain some further simplification in the proof.

As in [27], the proof is by induction on the dimension $m$. When $m = 1$, $A$, $f$, and $p$ are real numbers, with $A \geq 0$, and

$$\phi(p) = \frac{1}{2}Ap^2 + fp. \tag{62}$$

Since $\phi$ is bounded from below for all real $p$ (and not merely for $p \geq 0$), $A = 0 \Rightarrow f = 0$. Thus, if $A = 0$, $\phi(p) = 0$ for all $p$, and $p^\star$ can be any non-negative real number. If $A > 0$ and $f \geq 0$, then $p^\star = 0$ minimizes $\phi(p)$ over $p \geq 0$, and if $A > 0$ and $f < 0$, then $p^\star = -f/A > 0$ minimizes $\phi(p)$ over all real $p$, so the existence of $p^\star$ is proved for $m = 1$.

Now we make the induction hypothesis that the existence part of claim 1 is valid for all $m < M$. Let $S_+^M$ be the intersection of the unit sphere in $\mathbb{R}^M$ with $\mathbb{R}_+^M$:

$$S_+^M = \{s \in \mathbb{R}^M : \| s \| = 1 \ \& \ s \geq 0\}, \tag{63}$$

where $\|\|$ denotes the Euclidean norm. Note that any $p \in \mathbb{R}_+^M$ can be written as

$$p = \alpha s, \ \ \alpha \geq 0, \ \ s \in S_+^M, \tag{64}$$

and then

$$\phi(p) = \phi(\alpha s) = \frac{1}{2}\alpha^2 s^T As + \alpha s^T f. \tag{65}$$

Since $A$ is positive semi-definite $s^T As \geq 0$. There are then two possibilities: $s^T As > 0$ for all $s \in S_+^M$, or $s^T As = 0$ for some $s \in S_+^M$. We consider these possibilities separately.

In the first case, since $s^T As$ is a positive and continuous function of $s$ on the compact set $S_+^M$, we have that $\exists s_0 \in S_+^M$ such that

$$0 < s_0^T As_0 \leq s^T As. \tag{66}$$

By the Schwarz inequality, we also have

$$|s^T f| \leq \| f \|, \tag{67}$$

for all $s \in S_+^M$. Therefore, for all $\alpha \geq 0$ and all $s \in S_+^M$,

$$\begin{aligned} \phi(\alpha s) &\geq \frac{\alpha^2}{2}s_0^T As_0 - \alpha \| f \| \\ &= \alpha\left(\frac{1}{2}\alpha s_0^T As_0 - \| f \|\right). \end{aligned} \tag{68}$$

Let

$$\alpha_0 = \frac{2 \, \| f \|}{s_0^{\mathrm{T}} A s_0}. \tag{69}$$

From (68), we then have

$$\alpha > \alpha_0 \Rightarrow \phi(\alpha s) > 0. \tag{70}$$

Now let

$$\phi_{\mathrm{inf}} = \inf\{\phi(p) : p \in \mathbb{R}_+^M\}. \tag{71}$$

Since $\phi(0) = 0$, $\phi_{\mathrm{inf}} \leq 0$, and moreover if $\phi_{\mathrm{inf}} = 0$ we may set $p^\star = 0$, and then the existence part of claim 1 is established. Thus, we only need to consider $\phi_{\mathrm{inf}} < 0$. By definition of inf, there is a sequence $\{\cdots \alpha_k s_k \cdots\}$ such that $\alpha_k \geq 0$, $s_k \in S_+^M$, and

$$\phi(\alpha_k s_k) \rightarrow \phi_{\mathrm{inf}} \tag{72}$$

as $k \rightarrow \infty$. Since we are here considering $\phi_{\mathrm{inf}} < 0$ and because of (70), only finitely many of the $\alpha_k$ can be greater than the constant $\alpha_0$ defined by (69). Discarding these finitely many points, we are left with a sequence of points of a closed, bounded, and therefore compact subset of $\mathbb{R}_+^M$, namely

$$\{\alpha s : 0 \leq \alpha \leq \alpha_0, s \in S_+^M\}. \tag{73}$$

Any sequence of points of a compact set has a convergent subsequence with its limit in that set, and we call the limit $p^\star$. Then, since $\phi$ is continuous

$$\phi(p^\star) = \phi_{\mathrm{inf}} \leq \phi(p), \tag{74}$$

for all $p \in \mathbb{R}_+^M$, so the existence of $p^\star$ in the in the case that $s^{\mathrm{T}} A s > 0$ for all $s \in S_+^M$ is proved.

It remains to consider the case in which $s^{\mathrm{T}} A s = 0$ for some $s \in S_+^M$. We will use such an $s$ to show that for any point $p$ in the interior of $\mathbb{R}_+^M$ there is some point $q$ on the boundary of $\mathbb{R}_+^M$ such that $\phi(p) = \phi(q)$.

Let $p$ be any point in the interior of $\mathbb{R}_+^M$. Such a point has

$$p_j > 0 \text{ for } j = 1, \ldots, M. \tag{75}$$

Then, since $s^{\mathrm{T}} A s = 0$,

$$\begin{aligned}
\phi(p + \lambda s) &= \frac{1}{2}(p + \lambda s)^{\mathrm{T}} A(p + \lambda s) + (p + \lambda s)^{\mathrm{T}} f \\
&= \phi(p) + \lambda (Ap + f)^{\mathrm{T}} s.
\end{aligned} \tag{76}$$

Since $\phi$ is bounded from below over all of $\mathbb{R}^M$ (and not merely over $\mathbb{R}_+^M$), the coefficient of $\lambda$ must be zero. Thus,

$$\phi(p + \lambda s) = \phi(p) \tag{77}$$

for all real $\lambda$.

For every $p$ in the interior of $\mathbb{R}_+^M$, and with $s$ fixed as above, the line $\{p + \lambda s : \lambda \in \mathbb{R}\}$ intersects the boundary of $\mathbb{R}_+^M$ in exactly one point. That point lies in at least one of the sets

$$B_j^M = \{p \in \mathbb{R}_+^M : p_j = 0\}. \tag{78}$$

It follows that the range of $\phi(p)$ for $p \in \mathbb{R}_+^M$ is the same as the range of $\phi(p)$ for $p \in \text{boundary}(\mathbb{R}_+^M)$. Each of the sets $B_j^M$ is isomorphic to $\mathbb{R}_+^{M-1}$, so the induction hypothesis is applicable to each of them. Thus, for each $j \in \{1, \ldots, M\}$, there is some $p^j \in B_j^M$ such that

$$\phi(p^j) \leq \phi(p) \tag{79}$$

for all $p \in B_j^M$. Also, since there are only a finite number of the $p^j$, there is at least one of them, say $p^\star$ such that

$$\phi(p^\star) \leq \phi(p^j), \ \ j = 1, \ldots, M. \tag{80}$$

Then (77–80) shows that $\phi(p^*) \leq \phi(p)$ for all $p \in \mathbb{R}_+^M$. This completes the existence part of claim 1.

Now we come to the part of claim 1 that concerns the possible non-uniqueness of $p^\star$. Suppose there is some $p^0 \in \mathbb{R}_+^M$ with $p^0 \neq p^\star$ such that

$$p^0 - p^\star \in \text{null}(A). \tag{81}$$

Then

$$\sum_{j=1}^m \pi_{ij}(p_j^0 - p_j^\star) = 0, \tag{82}$$

and of course this is equivalent to

$$\sum_{j=1}^m \pi_{ij} p_j^0 = \sum_{j=1}^m \pi_{ij} p_j^\star, \tag{83}$$

which implies that

$$\phi(p^0) = \phi(p^\star), \tag{84}$$

see (57). Thus, if $p^\star \in \mathbb{R}_+^m$ minimizes $\phi$ over $\mathbb{R}_+^m$ and if $p^0 \in \mathbb{R}_+^m$ differs from $p^\star$ by a vector in the null space of $A$, then $p^0$ is also a minimizer of $\phi$ over $\mathbb{R}_+^m$.

To complete the proof of claim 1, we need to prove the converse of the foregoing, i.e., that any two possibly different points of $\mathbb{R}_+^m$ that are both minimizers of $\phi$ over $R_+^m$ differ by a vector in the null space of $A$.

Let $p^\star$ and $p^0$ be two points of $\mathbb{R}_+^m$, both of which are minimizers of $\phi$ over $\mathbb{R}_+^m$. Then

$$\phi(p^\star) \leq \phi(p) \ \text{ and } \ \phi(p^0) \leq \phi(p) \tag{85}$$

for all $p \in \mathbb{R}_+^m$. An obvious consequence of (85) is that

$$\phi(p^\star) = \phi(p^0) \tag{86}$$

Since $\phi$ is convex,

$$\phi(p^0) \geq \phi(p^\star) + \sum_{j=1}^m \left(p_j^0 - p_j^\star\right) \frac{\partial \phi}{\partial p_j}(p^\star), \tag{87}$$

and because of (86), this is equivalent to

$$0 \geq \sum_{j=1}^{m} \left( p_j^0 - p_j^\star \right) \frac{\partial \phi}{\partial p_j} (p^\star), \tag{88}$$

Since $p^\star$ minimizes $\phi$ over $\mathbb{R}_+^m$,

$$\frac{\partial \phi}{\partial p_j} (p^\star) \geq 0, \tag{89}$$

with equality if $p_j^\star > 0$. Thus, every term in the sum on the right-hand side of (88) is non-negative, since every term with $p_j^\star > 0$ has $\frac{\partial \phi}{\partial p_j}(p^\star) = 0$, and every term with $p_j^\star = 0$ satisfies both (89) and $p_j^0 - p_j^\star = p_j^0 \geq 0$. Thus (88) states that a sum of non-negative terms is non-positive, and this is only possible if every term, and therefore the whole sum, is equal to zero. Therefore,

$$\begin{aligned} 0 &= \sum_{j=1}^{m} \left( p_j^0 - p_j^\star \right) \frac{\partial \phi}{\partial p_j} (p^\star) \\ &= (p^0 - p^\star)^{\mathrm{T}} (A p^\star + f). \end{aligned} \tag{90}$$

Now reversing the roles of $p^\star$ and $p^0$, we also have

$$0 = (p^\star - p^0)^{\mathrm{T}} (A p^0 + f), \tag{91}$$

and by adding (90) and (91), we get the result

$$0 = -(p^\star - p^0)^{\mathrm{T}} A (p^\star - p^0), \tag{92}$$

from which it follows that

$$\sum_{j=1}^{m} \pi_{ij} (p_j^\star - p_j^0) = 0, \tag{93}$$

so $(p - p^0)$ is indeed in the null space of A. (More generally, since A is symmetric and positive semi-definite, (92) implies that $p^\star - p^0$ is in the null space of $A$, as can be shown by expanding $p^\star - p^0$ in terms of the eigenvectors of $A$.) This completes the proof of claim 1.

The proof of claim 2, which is the unconstrained version of claim 1, is simpler. Note that the gradient of $\phi$ is $Ap + f$. Therefore, if $p$ minimizes $\phi$ over $\mathbb{R}^m$,

$$Ap + f = 0. \tag{94}$$

Moreover, since $\phi$ is convex, any $p$ that satisfies (94) minimizes $\phi$ over $\mathbb{R}^m$. The proof of this is very simple. If $p$, $p'$ are any two points of $\mathbb{R}^m$, the convexity of $\phi$ implies that

$$\phi(p') \geq \phi(p) + (p' - p)^{\mathrm{T}} (\mathrm{grad}\phi)(p). \tag{95}$$

Now if $p$ is such that $(\mathrm{grad}\phi)(p) = 0$, this becomes simply $\phi(p') \geq \phi(p)$ for all $p' \in \mathbb{R}^m$, so $p$ is indeed a minimizer of $\phi$.

Since $A$ is symmetric, (94) has at least one solution if $f$ is orthogonal to the null space of $A$, and this is indeed the case, see (56) and (60). This completes the proof of the existence part of claim 2.

**Table 1. Convergence test under PE1.**

| Time step | Maximum difference in log($r$) | Ratio with coarser result |
|-----------|-------------------------------|---------------------------|
| $\Delta t$ and $\Delta t/2$ | 0.0093 | 2.0020 |
| $\Delta t/2$ and $\Delta t/4$ | 0.0047 | |

Now let $p^{\star\star}$ minimize $\phi$ over $\mathbb{R}^m$, and let $p^{00} - p^{\star\star}$ be in the null space of $A$. Then

$$Ap^{\star\star} + f = 0, \tag{96}$$

$$A(p^{00} - p^{\star\star}) = 0, \tag{97}$$

and it follows that

$$Ap^{00} + f = 0. \tag{98}$$

Eq (98) implies that $p^{00}$ minimizes $\phi$ over $\mathbb{R}^m$, since $Ap + f$ is the gradient of $\phi$ and since $\phi$ is convex (see argument following Eq (94)).

Conversely, if $p^{\star\star}$ and $p^{00}$ both minimize $\phi$ over $\mathbb{R}^m$, then Eqs (96) and (98) are both satisfied, and these together imply (97), so $p^{00} - p^{\star\star}$ is in the null space of $A$. This completes the proof of claim 2.

Note that non-uniqueness, if any, of the price vector $p$ is restricted in the same way in the constrained and unconstrained minimization of $\phi(p)$. In both cases, any two solutions of the minimization problem differ at most by a vector $q$ that satisfies $\sum_{j=1}^{m} \pi_{ij} q_j = 0$ for $i = 1, \ldots, n$. As is clear from Eqs (6) and (18 and 19) of the main text, such non-uniqueness has no effect on the evolution of the model economy.

## Appendix 2

We discuss the results of the convergence test in this appendix. A convergence study has been performed on one of the 30 pairs of simulations. For each price-equilibrium scenario, the simulation has been run with identical parameters (that of $\pi$, $c$ and initial $r$) three times with different time steps—that of the original $\Delta t$, $\Delta t/2$, and $\Delta t/4$. The three simulations are run for the same length of time, with the second and third having twice and four times the number of iterations of the original. When calculating the maximum difference between two simulations, we take into account the significant spread of the r values in later time, and record maximum absolute value of difference between the log of the $r$ value of all respective process across time. It can be seen from the results presented in Tables 1 and 2, that we can observe a linear convergence under both price equilibrium schemes. A graphical representation is not included as the results gained from each simulation are very close to each other and can not be easily distinguished on a graph.

**Table 2. Convergence test under PE2.**

| Time step | Maximum difference in log($r$) | Ratio with coarser result |
|-----------|-------------------------------|---------------------------|
| $\Delta t$ and $\Delta t/2$ | 0.0106 | 2.0021 |
| $\Delta t/2$ and $\Delta t/4$ | 0.0053 | |

## Supporting information

**S1 Graphical abstract.**
(PDF)

## Author Contributions

**Conceptualization:** Shubo (Gabriel) Xu, Charles S. Peskin.

**Data curation:** Shubo (Gabriel) Xu.

**Formal analysis:** Shubo (Gabriel) Xu, Charles S. Peskin.

**Investigation:** Charles S. Peskin.

**Methodology:** Charles S. Peskin.

**Project administration:** Charles S. Peskin.

**Software:** Shubo (Gabriel) Xu.

**Supervision:** Charles S. Peskin.

**Visualization:** Shubo (Gabriel) Xu.

**Writing – original draft:** Shubo (Gabriel) Xu, Charles S. Peskin.

**Writing – review & editing:** Shubo (Gabriel) Xu, Charles S. Peskin.

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
