## [Decision Letter · Decision Letter 0]

29 Sep 2021

PONE-D-21-28836The Impact of Universal Recycling on the Evolution of Economic DiversityPLOS ONE

Dear Dr. Xu,

Thank you for submitting your manuscript to PLOS ONE. After careful consideration, we feel that it has merit but does not fully meet PLOS ONE’s publication criteria as it currently stands. Therefore, we invite you to submit a revised version of the manuscript that addresses the points raised during the review process. Please submit your revised manuscript by Nov 13 2021 11:59PM. If you will need more time than this to complete your revisions, please reply to this message or contact the journal office at plosone@plos.org. Please include the following items when submitting your revised manuscript:A rebuttal letter that responds to each point raised by the academic editor and reviewer(s). You should upload this letter as a separate file labeled 'Response to Reviewers'.A marked-up copy of your manuscript that highlights changes made to the original version. You should upload this as a separate file labeled 'Revised Manuscript with Track Changes'.An unmarked version of your revised paper without tracked changes. You should upload this as a separate file labeled 'Manuscript'.

We look forward to receiving your revised manuscript.

Kind regards,

Maria Alessandra Ragusa, PhD Professor

Academic Editor

PLOS ONE

Journal Requirements:

3. Your abstract cannot contain citations. Please only include citations in the body text of the manuscript, and ensure that they remain in ascending numerical order on first mention.

Additional Editor Comments:

Dear corresponding author,

the paper needs a major revision. Please, make all the changes according to it in a week and send it back.

Best regards

Reviewers' comments:

Reviewer's Responses to Questions

**Comments to the Author**

1. Is the manuscript technically sound, and do the data support the conclusions?

Reviewer #1: Yes

2. Has the statistical analysis been performed appropriately and rigorously? 

Reviewer #1: Yes

3. Have the authors made all data underlying the findings in their manuscript fully available?

Reviewer #1: No

4. Is the manuscript presented in an intelligible fashion and written in standard English?

Reviewer #1: Yes

5. Review Comments to the Author

Reviewer #1: In this work, the authors used numerical simulation to investigate its dynamics. At each timestep of the numerical method, there is a price-equilibrium problem to be solved, and they proved the existence and essential uniqueness of the solution under both versions of the model. What they observed, but do not prove, is the evolution into a balancedgrowth solution. During this evolution, prices gradually stabilize, many processes are gradually weeded out of of the evolving economy as the surviving processes become better and better tuned to each other, and the rate of economic growth gradually increases, eventually settling into a plateau in which the rate of economic growth is constant. This happens in both versions of the model, but with the following differences: In the first scenario, with prices constrained to be nonnegative and with excess supply allowed, the final rate of economic growth is higher, but the economy is more monopolistic, with fewer processes holding a greater percentage of capital. In the second scenario, with perfect market clearing enforced and negative prices allowed, the plateau of economic growth is reached at a lower level, and the mature economy is more diverse. These qualitative results are very robust, in the sense that they persist despite different choices of the parameters that define the model economy.

After carefully reviewed the paper, I conclude that:

* It seems that the study Paper is interesting.

* It seems that all results are mathematically correct.

* Novelty of the work also good.

But, I can accept the for publication after considering the following major revisions as follows:

(1) Main advantages of the finding are fully missing in the introduction?

(2) Convergence condition of all results are missing?.

(3) Reference section must be checked and revised according to the journal template.

(4) The references are very old.

(5) Why the present study is more important compared to other following recent works which deals the numerical analysis and can the author apply the given algorithm in his/her study in order to ensure the stability analysis?!

https://www.sciencedirect.com/science/article/abs/pii/S0096300311000415

https://onlinelibrary.wiley.com/doi/10.1002/mma.4525

https://onlinelibrary.wiley.com/doi/10.1002/mma.4306

https://www.sciencedirect.com/science/article/abs/pii/S0252960218308105

https://www.sciencedirect.com/science/article/pii/S0019357712000626

https://www.sciencedirect.com/science/article/abs/pii/S0096300315004099

https://www.sciencedirect.com/science/article/abs/pii/S0096300311000415

6. PLOS authors have the option to publish the peer review history of their article (what does this mean?). If published, this will include your full peer review and any attached files.

Reviewer #1: No

---

## [Author Response · Author response to Decision Letter 0]

24 Nov 2021

To the Editor and Reviewer:

We have revised our paper, ”The Impact of Universal Recycling on the Evolution of Economic Diversity,” based on the feedback of the Editor and the Reviewer. We have also uploaded all codes and data needed for the replication of the simulations in our study to an online repository recommended by the Journal. These codes and data can be accessed at https://doi.org/10.17026/dans-z3k-95ya. (The link is pending activation at this time, and should be accessible soon.)

We thank the Reviewer for kind remarks and helpful comments, which we think have resulted in improvements to the paper. Our responses to the specific comments are as follows:

1. Main advantages of the finding are fully missing in the introduction?

We interpret this comment to mean that the significance of our results are not dis- cussed in the Introduction. To remedy this, we now discuss in the revised Introduction the real-world implications of our findings.

In particular, we point out a connection between our work and a current business phenomenon called Upcycling, which is the concept of transforming unwanted or waste products of production into market-priced inputs to the productions of other goods. We return to this theme in the Discussion section, where we point out in more detail how the phenomenon of Upcycling is related to the predictions of our model.

In case the Reviewer’s comment was related to methodological contributions, we would just like to point out that these were already thoroughly discussed in the Introduction. These contributions include the addition of a capital component to von Neumann’s classic model which allows transformation of the discrete-time model into a continuous-time dynamic model described by system of ODEs; the way prices are determined without the need for the notion of utility; the straightforward computational methods we have implemented; as well as the diversity index we have created based on entropy in the distribution of capital.

2. Convergence condition of all results are missing?

We have added Appendix 2, in which we report the results of a convergence study on one of the 30 pairs of simulations that are described in the paper. Under each of the two price-equilibrium scenarios, simulations are run for the same length of time for three different time-step durations. The results are consistent with first- order convergence, as expected for Euler’s method, under both price-equilibrium scenarios. When we plot the more refined results on top of the coarser results, the results obtained at different levels of refinement are indistinguishable, which is why we have not included such graphs in the paper.

3. Reference section must be checked and revised according to the journal template.

The revised manuscript including the reference section has been formatted according to the PLOS ONE template.

4. The references are very old.

New references have been added, especially in relation to the recent phenomenon of Upcycling, the emergence of which is consistent with the predictions of our model. Since this phenomenon is very recent, the references are to news stories, and we now discuss the relationship between these accounts and the theory developed in our pa- per.

It is true that some references are old, but they are fundamental to our work. The classic von Neumann’s model is the foundation on which we build, and Walras’ Law is also of theoretical significance as we set up our model. We have compared our work to economic growth theories based on the prevalent neoclassical models, as well as other extensions of von Neumann’s model. There are recent works in both categories that we have discussed and referenced (see Discussion).

5. Why the present study is more important compared to other following recent works which deals the numerical analysis and can the author apply the given algorithm in his/her study in order to ensure the stability analysis?

It is true that we do not prove the stability of our scheme, but the computed results show that it is stable within the range of parameters that we actually use. This statement is reinforced by the empirical convergence study that we have added to the paper (see Appendix 2) in response to the Reviewer’s comment #2.

The papers cited by the Reviewer are concerned with semi-implicit time-differencing schemes for partial differential equations. Here, we are considering systems of ordinary differential equations, and we find that the explicit Euler’s method is adequate for our purposes. The use of this simple method has a conceptual advantage, already pointed out in the original version of the paper, that the formulae for excess demand are exactly the same whether they are derived before or after discretization.

More sophisticated numerical methods could certainly be used, and might be more efficient, but the results would be substantially the same. This statement, like the one made above about stability, is now reinforced by the empirical convergence study (Appendix 2), which indicates that the results are converging as the time step is re- fined.

With thanks for your continued consideration of this work, 

Sincerely yours,

Shubo Xu

Graduate Student in Scientific Computing

Charles S. Peskin

Silver Professor of Mathematics

---

## [Editor Report · Decision Letter 1]

19 Dec 2021

The Impact of Universal Recycling on the Evolution of Economic Diversity

PONE-D-21-28836R1

Dear Dr. Xu,

We’re pleased to inform you that your manuscript has been judged scientifically suitable for publication and will be formally accepted for publication once it meets all outstanding technical requirements.

Kind regards,

Maria Alessandra Ragusa, PhD Professor

Academic Editor

PLOS ONE

Additional Editor Comments (optional):

Dear corresponding Author,

the paper received the referee's comments.

Plase, do carefully the major revision request and send back the paper for final decision.

Best regards.
---

## [Editor Report · Acceptance letter]

5 Jan 2022

PONE-D-21-28836R1 

The impact of universal recycling on the evolution of economic diversity 

Dear Dr. Xu:

I'm pleased to inform you that your manuscript has been deemed suitable for publication in PLOS ONE. Congratulations! Your manuscript is now with our production department. 

Kind regards, 

on behalf of

Dr. Maria Alessandra Ragusa 

Academic Editor

PLOS ONE